# Conservation of heat stress acclimation by the IPK2-type kinases that control the synthesis of the inositol pyrophosphate 4/6-InsP$_7$ in land plants

Ranjana Yadav[1☙], Guizhen Liu[2☙], Priyanshi Rana[1], Naga Jyothi Pullagurla[1], Danye Qiu[2], Henning J. Jessen[2], Debabrata Laha[1*]

1 Department of Biochemistry, Division of Biological Sciences, Indian Institute of Science (IISc), Bengaluru, Karnataka, India, 2 Institute of Organic Chemistry & CIBSS - Centre for Integrative Biological Signalling Studies, University of Freiburg, Freiburg, Germany

☙ These authors contributed equally to this work.

* dlaha@iisc.ac.in

## Abstract

Inositol pyrophosphates (PP-InsPs) are soluble cellular messengers that integrate environmental cues to induce adaptive responses in eukaryotes. In plants, the biological functions of various PP-InsP species are poorly understood, largely due to the absence of canonical enzymes found in other eukaryotes. The recent identification of a new PP-InsP isomer with yet unknown enantiomeric identity, 4/6-InsP$_7$ in the eudicot *Arabidopsis thaliana*, further highlights the intricate PP-InsP signalling network employed by plants. Yet, the abundance of 4/6-InsP$_7$ in land plants, the enzyme(s) responsible for its synthesis, and the physiological functions of this species are all currently unknown. In this study, we show that 4/6-InsP$_7$ is ubiquitous in the studied land plants. Our findings demonstrate that the *Arabidopsis* inositol polyphosphate multikinase (IPMK) homologs, AtIPK2α and AtIPK2β phosphorylates InsP$_6$ to generate 4/6-InsP$_7$ as the predominant PP-InsP species *in vitro*. Consistent with this, AtIPK2α and AtIPK2β act redundantly to control 4/6-InsP$_7$ production *in planta*. Notably, activity of these IPK2 proteins is critical for heat stress acclimation in *Arabidopsis*. Our parallel investigations using the liverwort *Marchantia polymorpha* suggest that the PP-InsP synthase activity of IPK2 and role of IPK2 in regulating the heat stress response are conserved in land plants. Furthermore, we show that the transcription activity of heat shock factor (HSF) is regulated by IPK2 proteins, providing a mechanistic framework of IPK2-controlled heat stress tolerance in land plants. Collectively, our study indicates that IPK2-type kinases have played a critical role in transducing environmental cues for biological processes during land plant evolution.

**Data availability statement:** The authors confirm that all data underlying the findings are fully available without restriction. All relevant data are within the paper and its Supporting Information files.

**Funding:** This work is supported by the Department of Biotechnology (DBT) for grant no. BT/PR43116/BRB/10/2010/2021, and in part by the Anusandhan National Research Foundation (ANRF) SRG/2021/000951, the MoE-STARS/STARS-2/2023-0162, the Infosys Foundation, and the Indian Institute of Science start-up fund to DL. We are also thankful to the DST-FIST infrastructure fund. RY is supported by the IISc fellowship. PR is the recipient of the Prime Minister Research Fellowship (PMRF). NJP acknowledges Council of Scientific & Industrial Research (CSIR) for research fellowship. HJJ and GL acknowledge funding from the Volkswagen Foundation (VW Momentum Grant 98604) and DFG (JE 572/11-1). This study was supported by the German Research Foundation (DFG) under Germany's excellence strategy (CIBSS-EXC-2189-Project ID 390939984) to HJJ. The funders had no role in study design, data collection and analysis, decision to publish, or preparation of the manuscript.

**Competing interests:** The authors have declared that no competing interests exist.

## Author summary

Inositol pyrophosphates (PP-InsPs) are eukaryote-specific cellular messengers that control a plethora of critical physiological processes, ranging from cellular metabolism to cellular energetics, and nutrient sensing. The identification of a new inositol pyrophosphate species, $4/6\text{-InsP}_7$, suggests the presence of a more diverse PP-InsP signalling network in plants. To date, the molecular basis of $4/6\text{-InsP}_7$ production and its physiological function in plants remained completely elusive. We report the identification of a non-archetypal function of inositol polyphosphate multikinase (IPMK/IPK2) that catalyzes the synthesis of $4/6\text{-InsP}_7$, and the kinase activity is critical for controlling heat stress acclimation. Furthermore, we show that the role of IPK2 in generating $4/6\text{-InsP}_7$ and regulating heat stress response is conserved in the studied land plants.

## Introduction

Inositol phosphates (InsPs) are phosphate esters of *myo*-inositol (Ins) that are synthesized by the combinatorial phosphorylation of the hydroxyl group (-OH) of the *myo*-inositol ring. The fully phosphorylated inositol ring, $InsP_6$, also known as phytic acid, is one of the most abundant InsP species that controls diverse cellular processes and serves as a substrate for the cellular messengers, inositol pyrophosphates (PP-InsPs). Diphosphate-containing PP-InsPs are characterized by the presence of "high energy" phosphoanhydride bonds, with $InsP_7$ and $InsP_8$ being the most characterized species [1–8]. In yeast and metazoans, PP-InsPs serve as the critical cellular messengers controlling a large array of physiological processes including phosphate homeostasis [7,9–11], cellular energetics [12] and metabolism [13–15]. To date, the metabolic pathways leading to the production of PP-InsPs are well established in yeast, amoeba and metazoans [7,16–19]. In *Saccharomyces cerevisiae*, phospholipase C-dependent pathway generates $Ins(1,4,5)P_3$ [20]. The yeast IPMK homolog, Ipk2 phosphorylates $Ins(1,4,5)P_3$ sequentially at the 6-OH and 3-OH positions to generate $Ins(1,3,4,5,6)P_5$ that serves as a precursor for $InsP_6$ [21]. In agreement with this, the yeast *ipk2* knockout strains lack detectable levels of $InsP_6$ [21,22]. Subsequently, the yeast Kcs1/ mammalian IP6K -type proteins phosphorylate the C5 position of $InsP_6$ and $1\text{-InsP}_7$ to generate $5\text{-InsP}_7$, and $1,5\text{-InsP}_8$, respectively [16,23]. Furthermore, mammalian PPIP5K/yeast Vip1 kinases phosphorylate $InsP_6$ and $5\text{-InsP}_7$ to generate $1\text{-InsP}_7$ and $1,5\text{-InsP}_8$, respectively [17,18,24,25].

Notably, the metabolic pathway of PP-InsP production is partially conserved in plants [8,26–29]. For instance, plants lack the canonical Kcs1/IP6K-type proteins responsible for $5\text{-InsP}_7$ production found in yeast and metazoans. By contrast, *Arabidopsis* ITPK1 and ITPK2 that are not sequence related to yeast Kcs1 or mammalian IP6K enzymes, phosphorylate $InsP_6$ to generate $5\text{-InsP}_7$ *in vitro* [29–32] and *in planta* [33,34]. Notably, Vip1 isoforms could be identified in all available plant genomes [35–37]. *Arabidopsis* Vip1 isoforms, VIH1 and VIH2 that possess 1-kinase

activity, produce 1-InsP$_7$ and 1,5-insP$_8$ *in vitro* and contribute to InsP$_8$ synthesis *in planta* [33,35,36,38]. Thanks to these recent advances in understanding PP-InsP metabolism that we can now investigate the physiological functions of various PP-InsP species in plants. For instance, reduction in InsP$_8$ levels through perturbance of VIH2 function in *Arabidopsis* leads to the compromised immune response against insect herbivores and necrotrophic fungi [35,39]. Furthermore, VIH-derived PP-InsPs play crucial roles in regulating phosphate starvation responses [38,40–42]. *Chlamydomonas* Vip1 controls nutrient sensing [37]. Similarly, ITPK1 function is also implicated in phosphate homeostasis [33,43,44] and hormonal responses [34,45] in plants. Enzymes responsible for PP-InsP production are also linked to plant immunity against pathogenic bacteria in *Arabidopsis* [46]. Collectively, these studies reinforce that PP-InsPs are critical cellular messengers regulating various aspect of plant physiology, immunity and development.

Recent studies have reported the presence of a new PP-InsP isomer with yet unknown enantiomeric identity, 4/6-InsP$_7$ in *Arabidopsis* and *M. polymorpha* tissue extracts [33,34,44,47,48]. In the social amoeba *Dictyostelium discoideum*, 4/6-InsP$_7$ represents the most abundant InsP$_7$ species and is synthesized by the amoeba Ip6k [19]. Since, plants lack the canonical IP6K, it is unclear how 4/6-InsP$_7$ is produced in plant cells. Furthermore, whether 4/6-InsP$_7$ is specific to a certain plant lineage or is ubiquitous in land plants are yet to be explored. Consequently, physiological functions of this newly identified PP-InsP species have not been characterized.

In this study, we demonstrate that 4/6-InsP$_7$ is the predominant form of InsP$_7$ present in the studied land plant tissues. To identify the protein(s) responsible for 4/6-InsP$_7$ production, we performed a structural-based homology screening, where we identified *Arabidopsis* IPK2, a member of inositol polyphosphate multikinase (IPMK) family [21,22,49–54], as the primary hit. Using *in vitro* biochemical assays, we show that indeed both *Arabidopsis* AtIPK2α and AtIPK2β proteins phosphorylate InsP$_6$ to generate 4/6-InsP$_7$. Consistent with this finding, analyses of IPK2-deficient plants revealed that *Arabidopsis* IPK2 isoforms control the synthesis of 4/6-InsP$_7$ *in planta*. Furthermore, our analyses revealed that the activity of AtIPK2 proteins is critical for plant adaptation to heat stress. Our parallel investigations using the bryophyte species *M. polymorpha*, the first plant to conquer the land 500 million years ago [55], allow us to conclude that the functions of IPK2 in controlling cellular levels of 4/6-InsP$_7$ and facilitating heat stress acclimation are conserved in land plants.

## Results

### 4/6-InsP$_7$ is ubiquitous in the studied land plants

To investigate whether 4/6-InsP$_7$ is ubiquitous across land plants, we analyzed the inositol phosphate profile of different plant species representing diverse clades of the embryophytes. To this end, inositol phosphates from the 14-day-old gametophytic thallus of Tak-1 (male plant), Tak-2 (female plant) of *M. polymorpha* (Liverworts; Bryophyta), mature sporophylls of *Nephrolepis sp.* and *Dryopteris sp.* (Ferns; Pteridophyta) and 14-day-old seedlings of wild-type *Arabidopsis* (Eudicot; Angiosperms) were extracted and analyzed using capillary electrophoresis coupled with mass spectrometry method (CE-MS) [56–58]. Our analyses establish the presence of 4/6-InsP$_7$ in all the plant tissue extracts used in the study (Fig 1A). Furthermore, the quantification of PP-InsP species (in pmol/mg) revealed that the level of 4/6-InsP$_7$ was significantly higher than that of other InsP$_7$ isomers, i.e., 5-InsP$_7$ and 1/3-InsP$_7$ detected in the plant tissue extracts (Fig 1A). Altogether, these findings revealed that 4/6-InsP$_7$ is the predominant InsP$_7$ isomer detected in the specific tissue extracts of the embryophytes used in our study. Our CE-MS measurements further identified various InsP$_{3-4-5}$ species that are conserved in land plants (S1 Fig).

### Arabidopsis IPK2 proteins phosphorylate InsP$_6$ to generate 4/6-InsP$_7$ as the major PP-InsP species *in vitro*

We aimed to identify the protein(s) responsible for 4/6-InsP$_7$ production in plants (Fig 1B). Given that green plants lack canonical IP6K-type proteins as found, e.g., in mammals [16,59], we speculated that the plant genome might encode a novel InsP kinase that generates 4/6-InsP$_7$ from InsP$_6$. To test this hypothesis, we performed a homology search by Phyre$^2$ software [60] using human inositol hexakisphosphate kinase 3 (HsIP6K3) that converts InsP$_6$ to 5-InsP$_7$ [16,23],

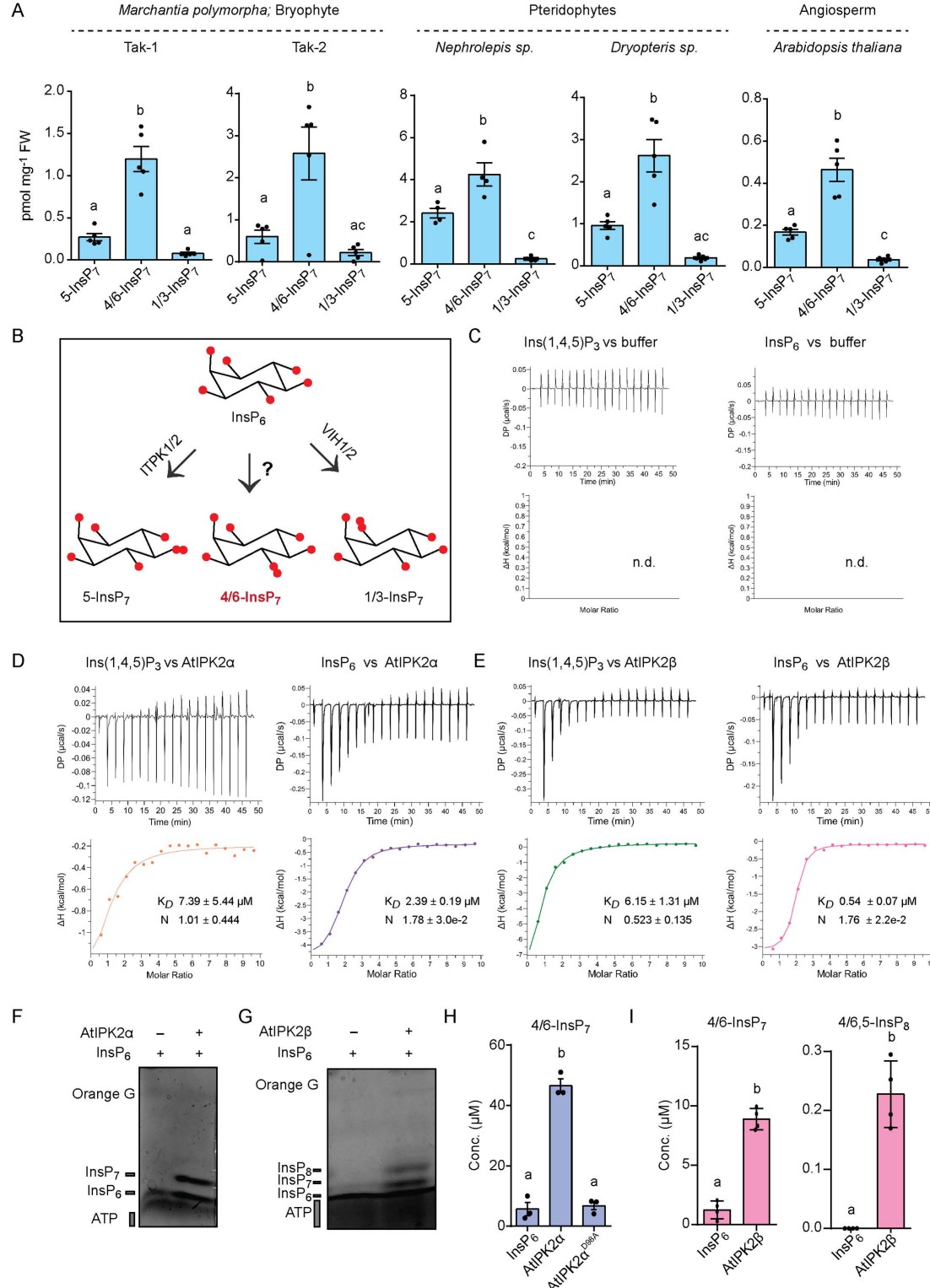

**Fig 1. 4/6-InsP$_7$ represents the major form of InsP$_7$ isomer in the studied land plant tissues and is synthesized by the Arabidopsis IPK2 proteins *in vitro*. A.** 4/6-InsP$_7$ is the major InsP$_7$ isomer detected in different embryophyte extracts. CE-MS analyses of InsP$_7$ isomers present in the extracts of the designated embryophytes. 5-InsP$_7$, 4/6-InsP$_7$ and 1/3-InsP$_7$ were assigned by mass spectrometry and identical migration time compared

with their heavy isotopic standards. Data are means±SE (n ≥ 4, biological replicates). Different letters indicate significance in one-way analysis of variance (ANOVA) followed by Tukey's test (a and b, $P < 0.0001$ for Tak-1; a and b, $P = 0.0079$, b and c, $P = 0.0023$ for Tak-2; a and b, $P = 0.011$, a and c, $P = 0.004$, b and c, $P < 0.0001$ for *Nephrolepis sp.*; a and b, $P = 0.0007$, b and c, $P < 0.0001$ for *Dryopteris sp.* and a and b, $P = 0.0001$, a and c, $P = 0.041$, b and c, $P < 0.0001$ for *Arabidopsis* Col-0). **B.** Schematic representation of PP-InsP metabolism in *A. thaliana*. $InsP_6$ is phosphorylated at position C5 by ITPK1/2 to form $5\text{-}InsP_7$. VIH1/2 phosphorylate $InsP_6$ at position C1 to produce $1/3\text{-}InsP_7$ *in vitro*. However, the protein(s) responsible for $4/6\text{-}InsP_7$ synthesis remained unidentified. **C.** Isothermal titration calorimetry (ITC) assays of $Ins(1,4,5)P_3$ (500 μM; left panel) and $InsP_6$ (500 μM; right panel) (in syringe) with ITC buffer (in cell), respectively. Raw heats per injection are shown in the top panel and the bottom panel represents the integrated heats of each injection (n.d., no detectable binding). **D and E.** AtIPK2α and AtIPK2β bind to $InsP_6$ with greater affinity than $Ins(1,4,5)P_3$ *in vitro*. Isothermal titration calorimetry (ITC) assays of $Ins(1,4,5)P_3$ (500 μM; left panel) and $InsP_6$ (500 μM; right panel) binding to AtIPK2α (10 μM), respectively (D). ITC assays of $Ins(1,4,5)P_3$ (500 μM; left panel) and $InsP_6$ (500 μM; right panel) binding to AtIPK2β (10 μM), respectively **(E)**. Raw heats per injection are shown in the top panel and the bottom panel represents the integrated heats of each injection, fitted to a one-site binding model (solid line). The insets show the dissociation constant ($K_D$) and binding stoichiometry (N) (±fitting error). **F.** AtIPK2α phosphorylates $InsP_6$ to produce $InsP_7$ isomer *in vitro*. PAGE analysis of the *in vitro* kinase reaction products of AtIPK2α. The reaction products were separated by 33% PAGE and visualized with toluidine blue. **G.** AtIPK2β phosphorylates $InsP_6$ to produce $InsP_7$ and $InsP_8$ *in vitro*. PAGE analysis of *in vitro* kinase reaction product of AtIPK2β. The reaction products were separated by 33% PAGE and visualized with toluidine blue. **H.** AtIPK2α synthesizes $4/6\text{-}InsP_7$ *in vitro*. CE-MS analyses of AtIPK2α-derived *in vitro* reaction products. Data represent means±SEM (n = 3, biological replicates). Different letters indicate significance in one-way analysis of variance (ANOVA) followed by Dunnett's test (a and b, $P < 0.0001$). The $InsP_7$ species was assigned by mass spectrometry and identical migration time compared with relative standards of $InsP_7$. **I.** AtIPK2β generates $4/6\text{-}InsP_7$ and $4/6,5\text{-}InsP_8$ *in vitro*. CE-MS analyses of AtIPK2β-derived *in vitro* reaction products. Data represent means±SEM (n = 4, replicates). Different letters indicate significance determined by Student's *t* test (a and b, $P < 0.0001$ for $4/6\text{-}InsP_7$; a and b, $P = 0.0002$ for $4/6,5\text{-}InsP_8$). $InsP_7$ and $InsP_8$ species were assigned by mass spectrometry and identical migration time compared with relative standards of $InsP_7$ and $InsP_8$.

as a query sequence to identify proteins having similar structural fold albeit poor sequence homology. Notably, the *Arabidopsis* inositol polyphosphate kinase alpha (AtIPK2α) was found as one of the primary hits (PDB ID: 4FRF) (S2 Table and S2A Fig) belonging to the inositol polyphosphate multikinase (IPMK) family [51,53]. Our analyses with structural models of HsIP6K3 and AtIPK2α revealed overall structural similarity between these two proteins, albeit poor sequence homology (S2A Fig). The *Arabidopsis* genome encodes two IPMK homologs, AtIPK2α and AtIPK2β [51,53,61]. We did not find AtIPK2β as a hit by the Phyre² search likely due to the lack of published crystal structure deposited in protein data bank portal. Similar to the yeast IPK2, both Arabidopsis IPK2 proteins phosphorylate $Ins(1,4,5)P_3$ and $Ins(1,4,5,6)P_4$ to produce $Ins(1,3,4,5,6)P_5$ *in vitro* [50,51,53]. Interestingly, *Arabidopsis* mutant seedlings defective in AtIPK2β function display a rather similar InsP profile compared to wild-type plants [50], suggesting that AtIPK2β might have different substrate specificities *in vivo* and that AtIPK2α and AtIPK2β may act redundantly to control InsP homeostasis. In contrast, previous studies had reported that AtIPK2β deficiency has a distinct effect on seed InsP metabolism than seedlings [36,50,62,63], suggesting that AtIPK2β executes specialized catalytic activity in different plant tissues. Given the catalytic flexibility of IPK2 members in recognizing different substrates such as $Ins(1,4,5)P_3$ and $Ins(1,4,5,6)P_4$ [21,22,51,53], we wondered whether Arabidopsis IPK2 proteins have evolved to possess PP-InsP synthase activity. The recent identification of an IP6K member that belongs to the IPMK family, responsible for the synthesis of $4/6\text{-}InsP_7$ in the social amoeba *Dictyostelium discoideum* [19], further encouraged us to explore the potential of AtIPK2 proteins to function as a $4/6\text{-}InsP_7$ synthase in plants.

To determine whether Arabidopsis IPK2 homologs recognize $InsP_6$ as a substrate, we tested the binding affinities of these proteins for $InsP_6$ under *in vitro* conditions. We purified tag-free Arabidopsis IPK2 proteins by subjecting recombinant AtIPK2 proteins having an N-terminal translational fusion with $His_8$-MBP-TEV to TEV protease followed by affinity-based purification (S2B and S2C Fig). Quantitative isothermal titration calorimetry (ITC)-binding assays were performed to determine dissociation constants ($K_D$) for $Ins(1,4,5)P_3$ (canonical substrate) and $InsP_6$ to AtIPK2. We performed control ITC assays including injection of InsP ligands into buffer and injection of buffer into enzymes (Figs 1C, S2D and S2E). Our ITC assays determined the dissociation constants ($K_D$) for $Ins(1,4,5)P_3$ and $InsP_6$ to AtIPK2α to be ~7 μM and ~2 μM, respectively (Fig 1D). Furthermore, our analyses revealed that $Ins(1,4,5)P_3$ and $InsP_6$ bound to AtIPK2β *in vitro* with the $K_D$ of ~6 μM and ~0.5 μM, respectively (Fig 1E), suggesting that $InsP_6$ could be a substrate for AtIPK2 proteins.

To explore the potential of AtIPK2α and AtIPK2β in phosphorylating InsP$_6$ to generate InsP$_7$ species, we incubated InsP$_6$ and ATP with the purified recombinant AtIPK2 proteins S2F and S2G Fig). Incubation of recombinant AtIPK2α with InsP$_6$, resulted a clear band that migrates slower than InsP$_6$ as resolved by highly concentrated polyacrylamide gel electrophoresis (PAGE) [64] (Figs 1F and S2H). Similarly, when InsP$_6$ was incubated with AtIPK2β, bands migrating slower than InsP$_6$ could be detected by PAGE (Fig 1G). In agreement with the previously published report [51], we found that the $K_m$ ATP for Ins(1,4,5)P$_3$ of both AtIPK2 proteins to be in micromolar range, 62–150 μM (S2I Fig; left panel). Subsequent kinetic parameter analysis revealed that the $K_m$ ATP for InsP$_6$ of both AtIPK2α and AtIPK2β to be 1.8 – 2.3 mM (S2I Fig; right panel), indicating that Ins(1,4,5)P$_3$ is a preferred *in vitro* substrate of AtIPK2. Intriguingly, these high $K_m$ ATP for InsP$_6$ values resemble the $K_m$ for ATP of mammalian IP6K1 [65,66] and Arabidopsis ITPK1 [29,33]. Additionally, the velocity of InsP$_6$ kinase reaction was surprisingly slow (S2I Fig), similar to what was previously observed for AtITPK1 proteins [29,33].

Next, the molecular identity of the reaction product was elucidated by CE-MS analyses, wherein the AtIPK2α and AtIPK2β-derived reaction products were spiked with corresponding heavy stable isotope labelled standards (SIL) [$^{13}$C$_6$]1-InsP$_7$, [$^{13}$C$_6$]5-InsP$_7$ [66] and [$^{18}$O$_2$]4-InsP$_7$ [67]. Our CE-MS analyses revealed that AtIPK2α and AtIPK2β-derived reaction products consist of different InsP$_7$ isomers, with 4/6-InsP$_7$ being the predominant species (Figs 1H, 1I, S2J and S2K). The precise enantiomeric identity of 4/6-InsP$_7$, i.e., 4-InsP$_7$ or 6-InsP$_7$ or a mixture of both species cannot be resolved yet using this conventional CE-MS method [68]. In addition to 4/6-InsP$_7$, 1/3-InsP$_7$ was also detected in the AtIPK2α-catalyzed reaction products (S2J Fig). Similarly, the CE-MS analyses of AtIPK2β-derived reaction products identified 4/6-InsP$_7$ as the major PP-InsP product (Figs 1I and S2K). Other PP-InsP isomers, including, 5-InsP$_7$, 1/3-InsP$_7$ and 4/6,5-InsP$_8$ could also be detected in the reaction products (Figs 1I and S2K). Time-course kinetic analyses further revealed that 4/6-InsP$_7$ is the predominant species produced by AtIPK2β *in vitro* (S2L Fig). Taken together, these data reflect that AtIPK2 proteins phosphorylate InsP$_6$ to generate different PP-InsP species *in vitro*, of which 4/6-InsP$_7$ is the predominant one.

Furthermore, to elucidate the role of the conserved PXXXDXKXG motif [51] of AtIPK2 proteins in its catalytic activity, translational fusion proteins of the catalytic dead mutants of AtIPK2α and AtIPK2β, i.e., AtIPK2α$^{D98A}$ and AtIPK2α$^{K100A}$ and AtIPK2β$^{D100A}$ and AtIPK2β$^{K102A}$ were generated (S2F and S2G Fig). Incubation of these catalytic dead mutants with InsP$_6$ and ATP did not result in more polar species than InsP$_6$ as determined by PAGE and CE-MS (Figs 1H, S2H, S2J and S2K), highlighting the importance of the key residues D98 and K100 for AtIPK2α and residues D100 and K102 for AtIPK2β in phosphorylating InsP$_6$. Collectively, our analyses identified a previously unreported non-canonical function of Arabidopsis IPK2 proteins as PP-InsP synthase, catalyzing the phosphorylation of InsP$_6$ to generate 4/6-InsP$_7$ *in vitro*.

## AtIPK2α and AtIPK2β act redundantly to control the synthesis of 4/6-InsP$_7$ *in planta*

Given the ability of Arabidopsis IPK2 isoforms to recognize various InsP substrates and to generate diverse InsP and PP-InsP species *in vitro*, we decided to examine carefully the InsP and PP-InsP profiles of AtIPK2-deficient plants using CE-MS to clarify the specific contribution of AtIPK2 proteins in regulating InsP and PP-InsP metabolism *in planta*. To this end, we studied the InsP profile of Col-0 (wild-type), *atipk2β-1* and *atipk2α-1* single knockout lines using CE-MS. The analysis revealed that *atipk2β-1* knockout plants exhibit a similar InsP profile when compared with the profile of Col-0 seedling extracts (Fig 2A). This is in agreement with the previously reported InsP profile of the same *atipk2β-1* T-DNA knockout line analyzed by SAX-HPLC [50]. Similar to the *atipk2β-1* mutant plants, the *atipk2α-1* single knockout seedlings also did not exhibit notable differences in the InsP profile compared to Col-0 plants, with the exception of significant accumulation of an InsP$_5$ isomer (Fig 2A). Additionally, we did not detect significant differences in any of the PP-InsP species between wild-type and the *atipk2* single knockout plants (Fig 2A). These results indicate that AtIPK2α and AtIPK2β act redundantly to control cellular InsP and PP-InsP metabolism. To corroborate further, we aimed to characterize *Arabidopsis* lines defective in both AtIPK2α and AtIPK2β. Notably, the *atipk2βatipk2α* double knockout plants are embryonic lethal [61], presenting a challenge for functional analysis of AtIPK2 homologs. To mitigate this limitation, we took an approach to generate

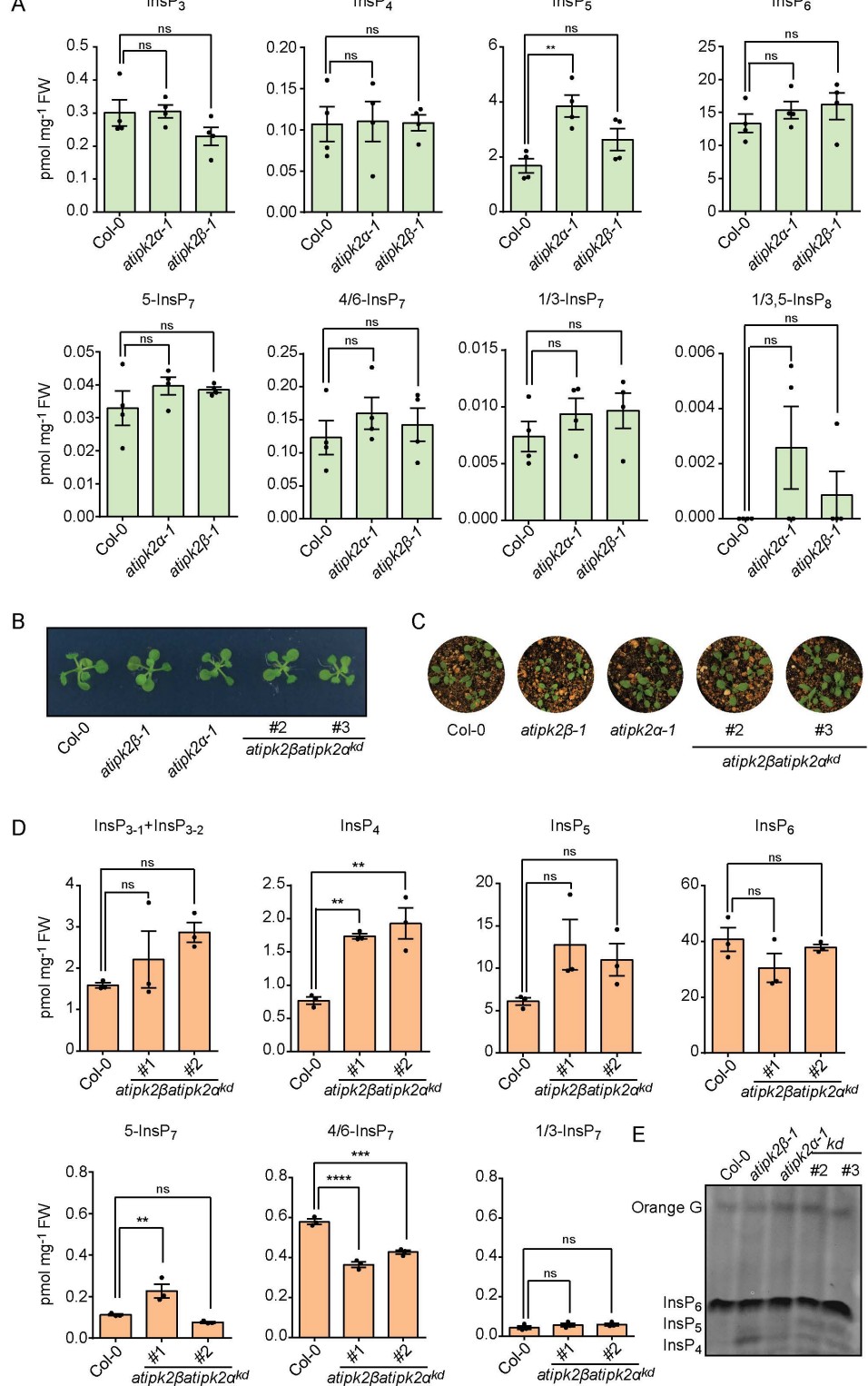

**Fig 2. Arabidopsis AtIPK2 α and AtIPK2 β regulate cellular level of 4/6-InsP₇.** **A.** CE-MS analyses of inositol phosphate extracts from shoot parts of 14-day-old Col-0, *atipk2α-1* and *atipk2β-1* seedlings. Values are ± SEM (n = 4, biological replicates). Statistical significance is determined in one-way ANOVA followed by Dunnett's test (**$P$ < 0.05). FW denotes fresh weight. **B.** Representative image of 14-day-old seedling of Col-0, *atipk2β-1, atipk2α-1*

and independent *atipk2βatipk2α^kd* lines grown on solidified half-strength MS media supplemented with 25 μM dexamethasone. Images were taken using Digital Single-Lens Reflex (DSLR) camera (Canon EOS 700D). **C.** Representative images of 4-week-old Col-0, *atipk2β-1, atipk2α-1* and *atipk2βatipk2α^kd* plants grown on perlite soil. Images were taken using a DSLR camera. **D.** AtIPK2-deficient plants exhibit deregulated levels of $InsP_4$ species and are compromised in $4/6\text{-}InsP_7$ production. CE-MS analyses of inositol polyphosphate extracts of 2-week-old Col-0 and *atipk2βatipk2α^kd* seedlings. The $InsP_5$, $InsP_6$ and $InsP_7$ species were assigned by mass spectrometry and identical migration time compared with their relative standards. One $InsP_4$ and two $InsP_3$ isomers were detected. Data are means±SEM (n = 3, biological replicates). Asterisk denotes significance determined in one-way ANOVA followed by Dunnett's test (**$P < 0.01$, *** $P < 0.001$, ****$P < 0.0001$). **E.** AtIPK2-deficiency does not affect global pool of $InsP_6$ production in *Arabidopsis* seedlings. PAGE analysis of inositol phosphates extracted from shoots of 14-day-old seedlings of Col-0, *atipk2β-1, atipk2α-1* and *atipk2βatipk2α^kd* lines grown on solidified half-strength MS media supplemented with 25 μM dexamethasone. Equal amount of plant tissues was used for the InsP extraction.

*Arabidopsis* lines with compromised expression of both *AtIPK2α* and *AtIPK2β*. We aimed to generate knockdown lines with reduced transcript levels of *AtIPK2α* in the *atipk2β-1* knockout plants. Following this strategy, we established stable lines for dexamethasone-inducible RNAi gene silencing of *AtIPK2α* in the *atipk2β-1* knockout plants by introduction of the complete coding DNA sequence (~900 bp) of *AtIPK2α* into the Hellsgate12 hairpin cassette under the dexamethasone-inducible bidirectional pOp6 promoter of the pOpOFF2(Hyg) vector [69,70] (S3A Fig). Selected independent knockdown (*kd*) lines were confirmed by genotyping (S3B Fig) and were tested for RNAi induction upon treatment with dexamethasone by means of a β-glucuronidase (GUS) reporter gene under control of the bidirectional pOp6 promoter (S3A and S3C Fig). Furthermore, the independent *atipk2βatipk2α^kd* knockdown lines showed compromised stability of *AtIPK2α* transcripts (S3D Fig). The *atipk2βatipk2α^kd* knockdown lines did not exhibit any obvious growth defects compared to the wild-type plants (Fig 2B and 2C). To illustrate the contribution of AtIPK2α and AtIPK2β in inositol phosphate homeostasis, we purified global InsP species from the extracts of respective genotypes using $TiO_2$ beads [71], and subjected the extracts to CE-MS analysis. The *atipk2βatipk2α^kd* seedlings exhibited an approximate two-fold increase in $InsP_4$ species of unknown isomeric identity and displayed no significant differences in $InsP_3$ and $InsP_6$ levels when compared with the wild-type plants (Fig 2D). This result was further supported by our PAGE analysis wherein all the genotypes had equal levels of $InsP_6$ and the *atipk2βatipk2α^kd* seedlings showed accumulation of unknown $InsP_4$ and $InsP_5$ isomers (Fig 2E). These findings are in agreement with the previously published report that AtIPK2 proteins can phosphorylate different lower InsP species [51,53] and further suggest that *Arabidopsis* IPK2 activity is not required for maintaining the global pool of $InsP_6$ in seedlings (Fig 2D and 2E). We would like to point out here that we were not able to detect any $InsP_7$ isomer in the *Arabidopsis* seedling extracts using PAGE (Fig 2E). Our CE-MS analyses revealed that the cellular levels of $1/3\text{-}InsP_7$ were not affected in the *atipk2βatipk2α^kd* seedlings (Fig 2D), suggesting that AtIPK2 proteins do not contribute to the production of $1/3\text{-}InsP_7$ species *in planta*. Although $5\text{-}InsP_7$ was accumulated in the *atipk2βatipk2α^kd* line #1, the knockdown line #2 did not show significant changes in this PP-InsP isomer, indicating that the changes in $5\text{-}InsP_7$ levels of the *atipk2βatipk2α^kd* line #1 may not be directly associated with AtIPK2 activity. Notably, the level of $4/6\text{-}InsP_7$ was significantly compromised in the independent knockdown lines compared to Col-0 plants (Fig 2D), suggesting that AtIPK2α and AtIPK2β cooperate together to regulate $4/6\text{-}InsP_7$ production *in planta.*

In agreement with previous observations [36,50,62,63], PAGE analyses of seed extracts confirmed that indeed seed $InsP_6$ levels are compromised in the *atipk2β-1* lines compared to wild-type plants (S4A Fig). Notably, our analysis further unveiled that AtIPK2α-defective seeds are also compromised in $InsP_6$ production (S4A Fig). Furthermore, PAGE analysis suggests that the seed phytate levels of the *atipk2βatipk2α^kd* lines are comparable to the single *atipk2* knockout plants. Taken together, these findings further highlight the distinct contribution of AtIPK2 isoforms in InsP metabolism between seed and seedlings.

## Heat stress specifically targets cellular levels of $4/6\text{-}InsP_7$

Given that the transcripts of *AtIPK2* isoforms could be altered during heat shock [72] (S3 Table), we sought to validate the transcriptomics data using qPCR analyses. We found that the transcript levels of both *AtIPK2α* and *AtIPK2β* are upregulated in response to heat stress (S4B Fig). Next, we asked whether heat stress influences inositol phosphate profile of the *atipk2βatipk2α^kd* lines compared to Col-0 plants. As depicted in Fig 3A, heat stress did not alter the $InsP_{3\text{-}4\text{-}5\text{-}6}$ levels of the

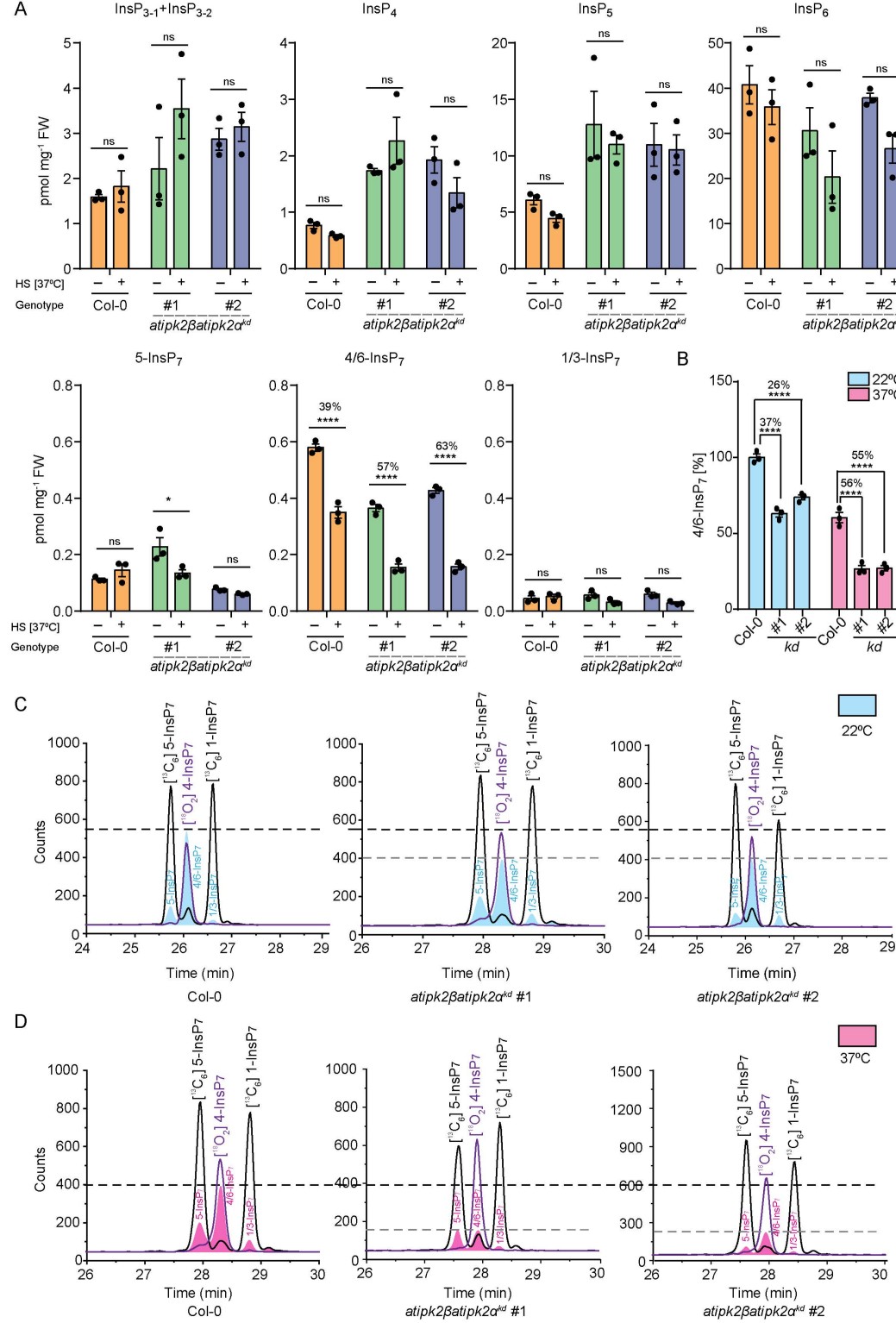

**Fig 3. Heat stress specifically targets 4/6-InsP$_7$. A.** AtIPK2-deficient plants exhibit compromised 4/6-InsP$_7$ production. CE-MS analyses of inositol polyphosphate levels of 2-week-old Col-0 and *atipk2βatipk2α$^{kd}$* seedlings. Seedlings were subjected to heat shock at 37ºC for 5 h and are harvested. Inositol phosphates were extracted by TiO2 pull down and were subjected to CE-MS analyses. The InsP$_5$, InsP$_6$ and InsP$_7$ species were assigned by

mass spectrometry and identical migration time compared with their relative standards. One InsP$_4$ and two InsP$_3$ isomers were detected. Data are means ± SEM (n = 3, biological replicates). Statistical significance is determined in two-way ANOVA followed by Tukey's test (*$P$ < 0.05, ****$P$ < 0.0001). **B.** Quantification of 4/6-InsP$_7$ in % of the data presented in (A). 4/6-InsP$_7$ level of wild-type plants grown under control condition was set to 100%. Data are means ± SE (n = 3, biological replicates). Statistical significance is determined in two-way ANOVA followed by Tukey's test (****$P$ < 0.0001). Data presented in Fig 2D served as the control group of Fig 3A and 3B. **C.** AtIPK2α and AtIPK2β regulate 4/6-InsP$_7$ production *in planta*. CE-MS analysis (extracted ion electropherograms) of InsP$_7$ in Col-0 and *atipk2βatipk2α$^{kd}$* lines (blue trace) with spiked [$^{13}C_6$] (black plot) and [$^{18}O_2$] (purple plot) labelled InsP$_7$. 4/6-InsP$_7$ was assigned by mass spectrometry and identical migration time compared with its heavy isotopic standards ([$^{18}O_2$] 4-InsP$_7$). **D.** CE-MS analysis (extracted ion electropherograms) of InsP$_7$ in Col-0 and *atipk2βatipk2α$^{kd}$* line after heat shock of 5 h at 37°C (pink trace) with spiked [$^{13}C_6$] (black plot) and [$^{18}O_2$] (purple plot) labelled InsP$_7$. 4/6-InsP$_7$ was assigned by mass spectrometry and identical migration time compared with its heavy isotopic standards ([$^{18}O_2$] 4-InsP$_7$). Note that Fig 3C and 3D are the representative CE-MS spectra of the experimental data presented in Fig 3A and 3B.

wild-type and the knockdown plants. Under control condition (22°C), the *atipk2βatipk2α$^{kd}$* lines showed 26–37% reduction in 4/6-InsP$_7$ level compared to the Col-0 plants (Fig 3B and 3C). Strikingly, the *atipk2βatipk2α$^{kd}$* lines suffered severely in 4/6-InsP$_7$ production with a ~56% reduction compared to the Col-0 plants when exposed to heat stress (37°C) (Fig 3A-3D). Notably, we did not observe significant changes in 1/3-InsP$_7$ levels in the AtIPK2-deficient plants during heat stress (Fig 3A), suggesting that AtIPK2 proteins do not contribute to 1/3-InsP$_7$ production *in planta*. Intriguingly, only one of the *atipk2βatipk2α$^{kd}$* lines exhibited significant downregulation of 5-InsP$_7$ levels (Fig 3A). Similar to 5 h heat treatment, the *atipk2βatipk2α$^{kd}$* lines displayed a robust reduction in 4/6-InsP$_7$ levels compared to the wild-type Col-0 plants when exposed to heat stress for 3 h (S4C Fig). Heat treatment did not affect other InsP$_7$ isomers in the *atipk2βatipk2α$^{kd}$* lines compared to Col-0 plants (S4C Fig).

## AtIPK2α and AtIPK2β cooperate to control heat stress acclimation in *Arabidopsis*

To elucidate the possible role of AtIPK2α and AtIPK2β in heat stress acclimation, we exposed Col-0, both the single knockout lines and the independent *atipk2βatipk2α$^{kd}$* lines to high ambient temperature and monitored the adaptive responses commonly referred as thermomorphogenesis [73–75]. Specifically, 7-day-old seedlings of the above-mentioned genotypes were grown either at 22°C or 28°C (high ambient temperature) up to five days and hypocotyl length was measured after 5 days. Expectedly, Col-0 showed heat-induced hypocotyl elongation (Fig 4A and 4B). Similar to Col-0, the *atipk2α-1* and *atipk2β-1* knockout plants also displayed heat-induced hypocotyl elongation. However, the *atipk2βatipk2α$^{kd}$* lines exhibited compromised hypocotyl elongation when grown under higher ambient temperature (Fig 4A and 4B), indicating that AtIPK2 contributes to shoot adaptive response to heat stress. To further interrogate the role of AtIPK2 isoforms in plant basal thermotolerance, defined as the ability of plant to survive under high temperature (37°C), we performed basal thermotolerance assays using Col-0, *atipk2β-1*, *atipk2α-1* and the three independent knockdown lines (Fig 4C-4E). Seedlings were initially grown at 22°C for 7 days in plant chamber. After this period, they were subjected to heat stress at 37°C for 3 days, followed by a recovery phase at 22°C for 4 days (Fig 4D). Plant survival was then assessed, defined by their ability to maintain, and generate fresh green leaves [76]. Notably, under control condition (22°C), the genotypes did not exhibit obvious difference in survival or germination (S5A Fig). However, the *atipk2βatipk2α$^{kd}$* lines were severely affected by heat stress showing poor survival rate as compared to Col-0 and the single mutant lines (Fig 4C and 4E). These data highlight the role of AtIPK2 isoforms in maintaining a plant's basal thermotolerance. Importantly, altered hypocotyl elongation during heat stress and compromised basal thermotolerance of the *atipk2βatipk2α$^{kd}$* plants were largely rescued by the expression of AtIPK2β under the control of a constitutive 35S promoter (Figs 4F-4J, S5B and S5C). Consistent with the role of AtIPK2 in heat stress acclimation, the independent *Arabidopsis* transgenic lines expressing *AtIPK2α* in translational fusion with a GFP tag under the control of a constitutive 35S promoter, exhibited increased hypocotyl length compared to their isogenic wild-type plants when exposed to heat stress (Figs 4K, 4L and S5D). In conclusion, our data suggest that AtIPK2α and AtIPK2β function redundantly to regulate heat stress acclimation in *Arabidopsis*.

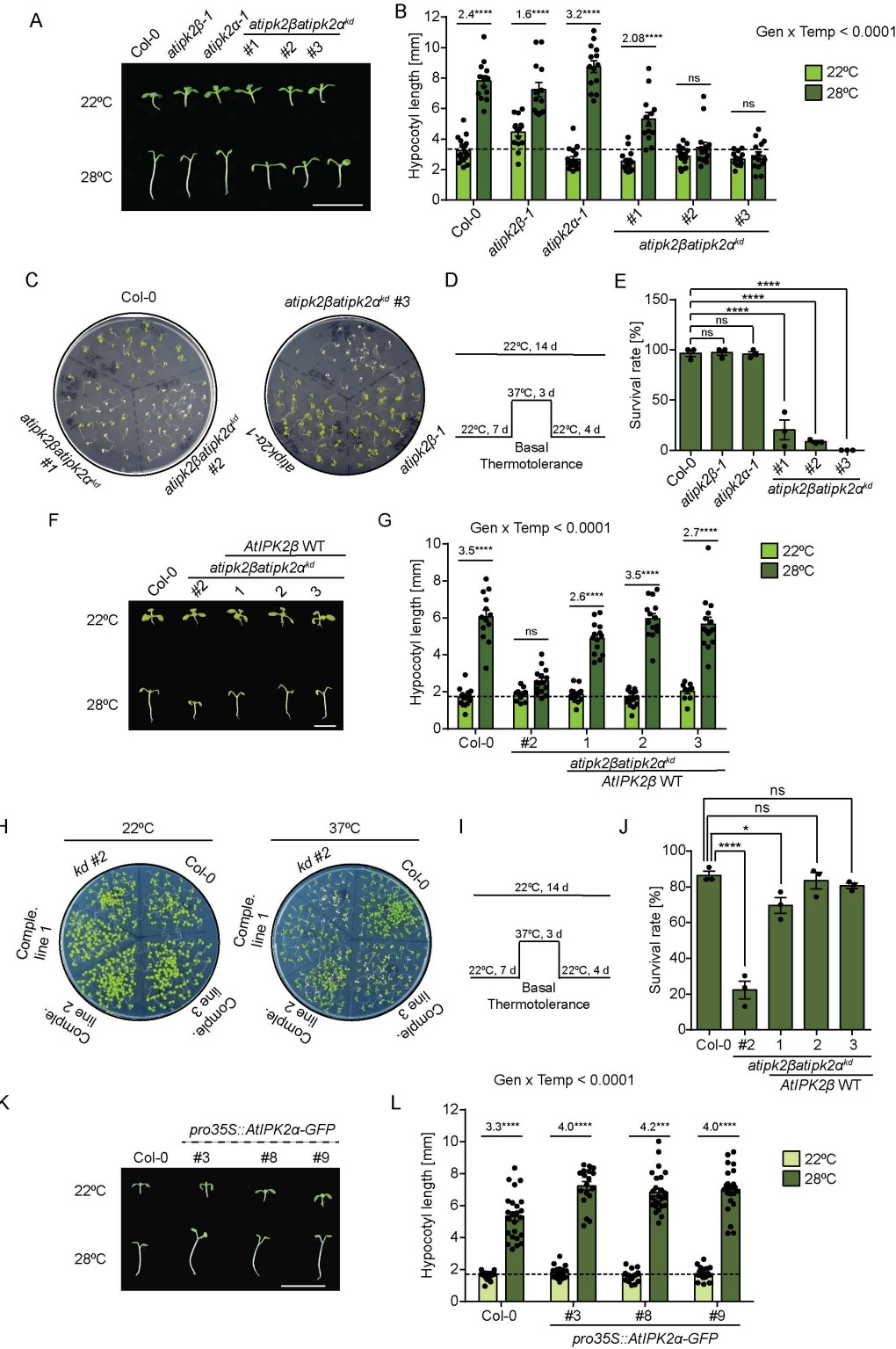

**Fig 4. AtIPK2α and AtIPK2β act redundantly to control heat stress acclimation in *Arabidopsis*. A.** AtIPK2α and AtIPK2β regulate hypocotyl elongation during heat stress. Representative photograph of high temperature-induced hypocotyl elongation phenotype of Col-0, *atipk2β-1*, *atipk2α-1* and *atipk2βatipk2α^kd* lines. 7-day-old seedlings were exposed to 28°C for 5 days. Control plates were maintained at 22°C. Images were captured after 5

days of heat stress. Scale bar = 1 cm. **B**. Quantification of high temperature-induced hypocotyl elongation of the designated genotypes grown at 22ºC and 28ºC. Hypocotyl elongation was evaluated by using ImageJ. Data are means ± SEM (n ≥ 13, biological replicates). Statistical significance is determined by two-way analysis of variance (ANOVA) followed by Tukey's test (**** $P < 0.0001$). Numbers on the bar represent the fold change in hypocotyl length upon heat stress compared to control condition in the respective genotypes. Dashed line represents the comparable hypocotyl length of designated genotype in control condition. **C**. Activity of Arabidopsis IPK2 isoforms is critical for survival during heat stress. Photograph showing basal thermotolerance phenotype of Col-0, atipk2β-1, atipk2α-1 and atipk2β atipk2α$^{kd}$ lines after heat stress at 37ºC. Surviving seedlings maintain green leaves and show emerged new leaves. **D**. Simplified experimental setup used for analysis of the survival phenotype. 7-day-old plants were exposed to 37ºC for 3 days and were subjected to subsequent recovery at 22ºC for 4 days. Control plates were maintained at 22ºC. **E**. Survival rate analysis of the designated genotypes at 37ºC. Values are means ±SEM, (n = 3, biological replicates) with each data point indicated and significant difference is determined by one-way analysis of variance (ANOVA) followed by Dunnett's test (**** $P < 0.0001$). The experiment was repeated several times with independent generation. **F**. Expression of AtIPK2β rescues attenuated hypocotyl elongation of the atipk2βatipk2α$^{kd}$ lines during heat stress. Representative photograph of high temperature-induced hypocotyl elongation phenotype of Col-0, atipk2βatipk2α$^{kd}$ lines and transgenic lines expressing AtIPK2β in the atipk2βatipk2α$^{kd}$ lines (Comple. lines 1/2/3). 7-day-old seedlings were exposed to 28°C for 5 days. Control plates were maintained at 22ºC. Images were captured after 5 days of heat stress. Scale bar = 6.5 mm. **G**. Quantification of high temperature-induced hypocotyl elongation of the designated genotypes grown at 22ºC and 28ºC. Hypocotyl elongation was evaluated by using ImageJ. Data are means ± SEM (n ≥ 10, biological replicates). Statistical significance is determined in two-way analysis of variance (ANOVA) followed by Tukey's test (**** $P < 0.0001$). Numbers on the bar represent the fold change in hypocotyl length upon heat stress compared to control condition in the respective genotypes. Dashed line represents the comparable hypocotyl length of designated genotype in control condition. **H**. Functional complementation of the AtIPK2-deficient plants compromised in heat stress acclimation by the expression of AtIPK2β. Photograph showing basal thermotolerance phenotype of Col-0, atipk2βatipk2α$^{kd}$ lines and transgenic lines expressing AtIPK2β in the atipk2βatipk2α$^{kd}$ lines after heat stress at 37ºC. Surviving seedlings maintain green leaves and show emerged new leaves. **I**. Simplified experimental setup used for analysis of the survival phenotype. 7-day-old plants were exposed to 37ºC for 3 days and were subjected to subsequent recovery at 22ºC for 4 days. Control plates were maintained at 22ºC. **J**. Survival rate analysis of the designated genotypes at 37ºC. Values are means ±SEM, (n = 3, biological replicates) with each data point indicated and significant difference is determined by one-way analysis of variance (ANOVA) followed by Dunnett's test (* $P < 0.05$, **** $P < 0.0001$). **K and L**. Overexpression of AtIPK2α leads to enhanced thermomorphogenetic response. Representative photograph of high temperature-induced hypocotyl elongation phenotype of Col-0 and three independent pro35S::AtIPK2α overexpression lines. 7-day-old seedlings were exposed to 28°C for 5 days. Control plates were maintained at 22ºC. Images were captured after 5 days of heat stress. Scale bar = 1 cm (K). Quantification of high temperature-induced hypocotyl elongation of the designated genotypes grown at 22ºC and 28ºC. Hypocotyl elongation was evaluated by using ImageJ. Data are means ± SEM (n ≥ 19, biological replicates). Statistical significance is determined in two-way analysis of variance (ANOVA) followed by Tukey's test (**** $P < 0.0001$) (L).

## AtIPK2 homologs are ubiquitous across plant kingdom and PP-InsP synthase activity of IPK2 is conserved in the liverwort *M. polymorpha*

Next, to explore the functional conservation of InsP kinase activity of AtIPK2 proteins in land plants, we constructed a phylogenetic tree including diverse taxa of plant kingdom such as green algae (Chlorophyta), liverworts and mosses (Bryophyta), lycopods (Pteridophyta), monocot and eudicot (Angiosperms) (S5 Table). The phylogenetic analysis allowed us to identify genes encoding IPK2-type proteins across the plant kingdom (S6A Fig). The analyses further suggest that these IPK2 homologs are derived from a single ancestral gene, with subsequent radiation in the individual lineages (S6A Fig). To understand the ancestral function of AtIPK2-type kinases in inositol phosphate homeostasis, we began to characterize the homolog of AtIPK2 in the liverwort *M. polymorpha*, a bryophyte whose genome sequence is available and is emerging as a model plant species to study land plant evolution (Fig 5A) [77,78]. The *M. polymorpha* genome encodes a single IPK2 homologue, named MpIPMK (as per nomenclature guidelines of *M. polymorpha*) [79] (S6A Fig).

The multiple sequence alignment of MpIPMK with AtIPK2α and yeast Ipk2 showed that MpIPMK possesses the conserved residues of the PXXXDXKXG catalytic motif suggesting that MpIPMK could be a functional homolog of AtIPK2 proteins (S6B and S7A Figs). Comparison of a structural model of MpIPMK with AtIPK2α further indicates that MpIPMK could be a functional IPK2-type InsP kinase (S7B-S7D Fig). We validated this hypothesis by taking advantage of the yeast (*S. cerevisiae*) ipk2Δ knockout strain and studied the consequences of heterologous expression of MpIPMK in these mutant strains. Ectopic expression of MpIPMK could restore the ipk2Δ-associated growth defects [51,53] at high temperature (37ºC) (Fig 5B). Rescue of the ipk2Δ-associated growth defects could only be noticed by the ectopic expression of wild-type MpIPMK but not by the catalytic dead mutants MpIPMK$^{D130A}$, MpIPMK$^{K132A}$ and MpIPMK$^{D130AK132A}$, indicating that MpIPMK possesses similar catalytic activity to yeast Ipk2 and AtIPK2α/β (Fig 5B). This conclusion was further substantiated by the HPLC analyses of the yeast ipk2Δ transformants expressing MpIPMK and MpIPMK$^{D130AK132A}$

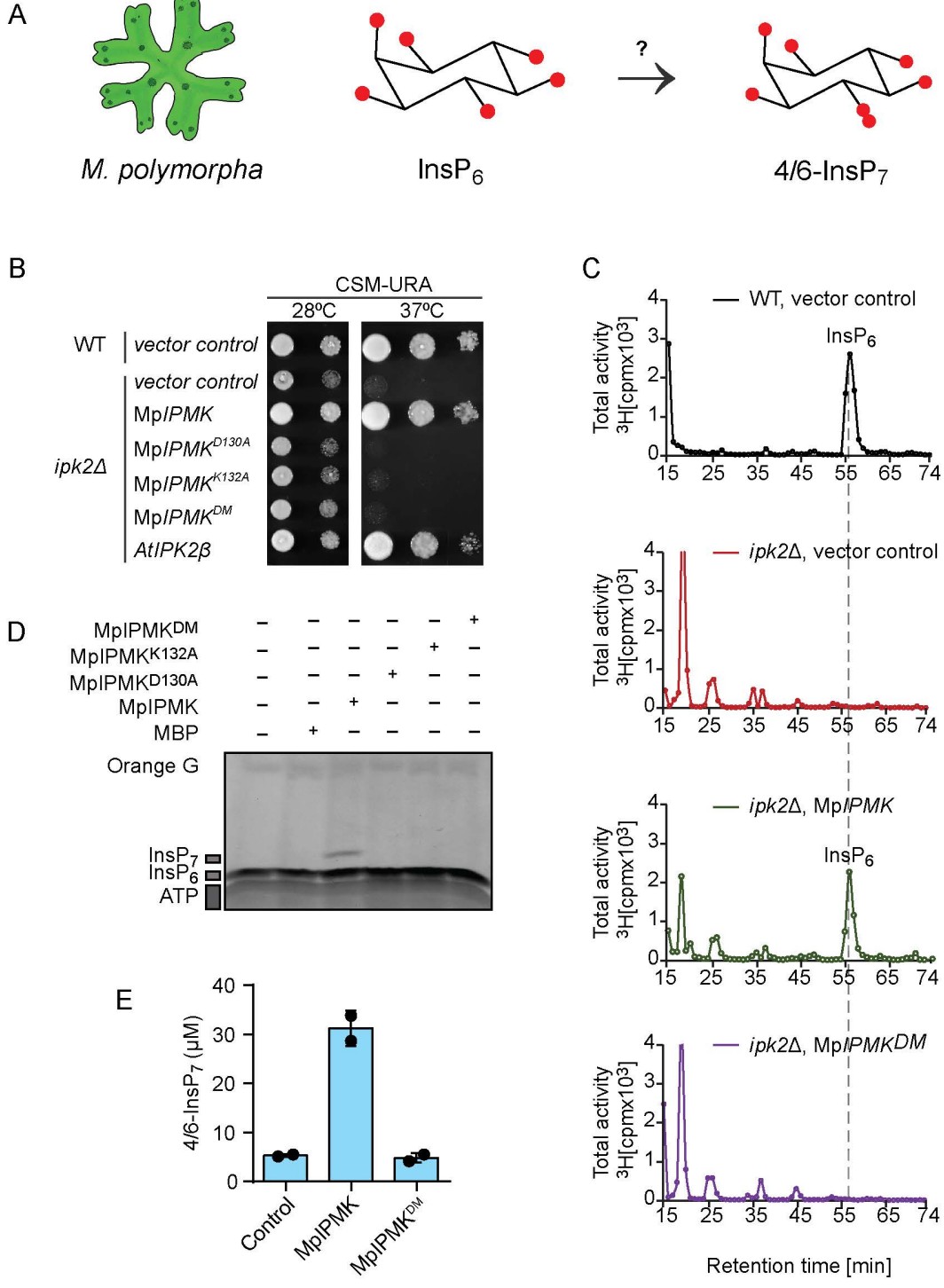

**Fig 5. *M. polymorpha* genome encodes a functional yeast Ipk2 homolog, MpIPMK that phosphorylates InsP$_6$ to generate PP-InsP isomers *in vitro*. A.** Schematic representation of 4/6-InsP$_7$ synthesis in *M. polymorpha*. **B.** MpIPMK is a functional yeast Ipk2 homolog. The *ipk2Δ* yeast strain transformed with episomal pCA45 (*URA3*) plasmid carrying Mp*IPMK* with N-terminal GST translation fusion were spotted in 8-fold serial dilution onto uracil-free plate and were incubated at 28ºC and 37ºC for 3 days. *AtIPK2β* served as positive control [51,53] and empty vector served as negative control. DDY1810 yeast strain was used for the experiment. Note: DM denotes MpIPMK$^{D130AK132A}$ catalytic dead variants. **C.** Complementation of the altered InsP profile of *ipk2Δ* yeast mutant by ectopic expression of Mp*IPMK*. HPLC profiles of extracts from [$^3$H]-*myo*-inositol-labelled yeast transformants.

Extracts were resolved by SAX-HPLC, and fractions collected each minute for subsequent determination of radioactivity as indicated. Experiments were repeated two times with similar results. **D.** MpIPMK phosphorylates $InsP_6$ *in vitro*. Recombinant $His_8$-MBP-MpIPMK and the catalytic dead proteins were incubated with 12.5 mM ATP, and 10 nmol $InsP_6$ at 37ºC for 12 h in reaction buffer. The reaction product was separated by 33% PAGE and visualized with toluidine blue. Experiments were repeated independent times with similar results. **E.** Production of 4/6-$InsP_7$ by MpIPMK *in vitro*. Quantification of MpIPMK-derived 4/6-$InsP_7$ using CE-MS. Data represent means ± SEM (n = 2, replicates). Experiments were repeated two times with similar results.

(Fig 5C). The altered InsP profile of the yeast *ipk2Δ* strain could be largely rescued by the ectopic expression of Mp*IPMK* (Fig 5C). In contrast, the catalytically dead variant of Mp*IPMK*, Mp*IPMK*<sup>D130AK132A</sup> failed to rescue the defective InsP profile (Fig 5C). These data allowed us to conclude that MpIPMK is a functional yeast Ipk2 homolog encoded by the *M. polymorpha* genome. To delineate whether MpIPMK possesses PP-InsP synthase activity similar to *Arabidopsis* IPK2 isoforms (Fig 5A), we incubated the translational fusion polypeptides of MpIPMK along with its catalytic dead variants (MpIPMK<sup>D130A</sup>, MpIPMK<sup>K132A</sup>, MpIPMK<sup>D130AK132A</sup>) with $InsP_6$ and ATP and the reaction products were resolved by PAGE (Figs 5D and S7E). The PAGE analyses revealed the presence of a clear kinase product when $InsP_6$ was incubated with the wild-type IPMK protein (Figs 5D and S7F) demonstrating that MpIPMK phosphorylates $InsP_6$ *in vitro*. The mutant MpIPMKs failed to synthesize more polar species of $InsP_6$ (Fig 5D). To get further insights into MpIPMK catalytic activity, we expressed Mp*IPMK* in different mutant yeast strains defective in PP-InsP metabolism. The expression of Mp*IPMK* did not rescue the *kcs1Δ*-associated growth defect (S8A Fig). Similarly, MpIPMK was not able to rescue the *vip1Δ*- associated growth defects (S8B Fig). Collectively, these data indicate that MpIPMK is a functional yeast Ipk2 homolog that neither possess Kcs1-type nor Vip1-type catalytic activity. To further corroborate the catalytic activity of MpIPMK and to deduce the structural identity of the MpIPMK reaction product, we subjected the *in vitro* MpIPMK-derivatives for CE-MS analyses. Notably, the MpIPMK reaction product showed exact comigration with the [$^{18}O_2$]4-$InsP_7$ standard, demonstrating that MpIPMK is the kinase responsible for 4/6-$InsP_7$ synthesis *in vitro* (Fig 5E). Similar to the AtIPK2 reaction products, a small amount of 1/3-$InsP_7$ could be detected using CE-MS analyses in the MpIPMK-derived reaction products (S8C Fig). Collectively, all these data unveil that MpIPMK is a functional IPK2-type InsP kinase that phosphorylates $InsP_6$ generating PP-InsP isomers *in vitro*.

### MpIPMK contributes to 4/6-$InsP_7$ synthesis *in planta* and controls heat stress responses

To decipher the contribution of MpIPMK in InsP and PP-InsP homeostasis, independent *M. polymorpha* knockout lines of Mp*IPMK* were generated using CRISPR-Cas9 technology [80,81] (Figs 6A and S9A). These mutant plants did not show any obvious growth defects compared to the wild-type plants (S9B Fig). To assess the consequence of altered expression of Mp*IPMK* in inositol phosphate metabolism, we monitored InsP profile of wild-type and Mp*ipmk* knockout plants using SAX-HPLC. As depicted in Fig 6B, the MpIPMK-deficient plants displayed accumulation of unknown $InsP_3$ isomer compared to the wild-type plants. Similar to the IPK2-defective *Arabidopsis* seedlings, MpIPMK deficiency did not affect the cellular $InsP_6$ levels of *M. polymorpha* thallus (Fig 6B). Given that SAX-HPLC offers limited information about structural identity of InsP isomers present in plant extracts, we performed CE-MS analyses of wild-type and the independent Mp*ipmk* knockout plant extracts. In congruence with the HPLC analysis, our CE-MS measurements revealed that the Mp*ipmk* knockout plants showed increased levels of different $InsP_3$ species (S9C Fig). One of the Mp*ipmk* knockout lines accumulated an $InsP_4$ isomer with unknown isomeric identity (S9C Fig). Notably, MpIPMK-defective plants were compromised significantly in their 4/6-$InsP_7$ level, suggesting that indeed MpIPMK contributes to 4/6-$InsP_7$ production *in planta* (Fig 6C). Given the role of AtIPK2α and AtIPK2β proteins in heat stress acclimation and that Mp*IPMK* transcript level is altered after exposure to heat stress (Fig 6D), we asked whether MpIPMK is involved in regulating PP-InsP metabolism during heat stress. Unlike *Arabidopsis* extracts, we were able to detect $InsP_7$ in the *M. polymorpha* extracts using PAGE, and thus, we employed PAGE analyses to monitor changes in cellular $InsP_7$ level during heat stress (Fig 6E). We found

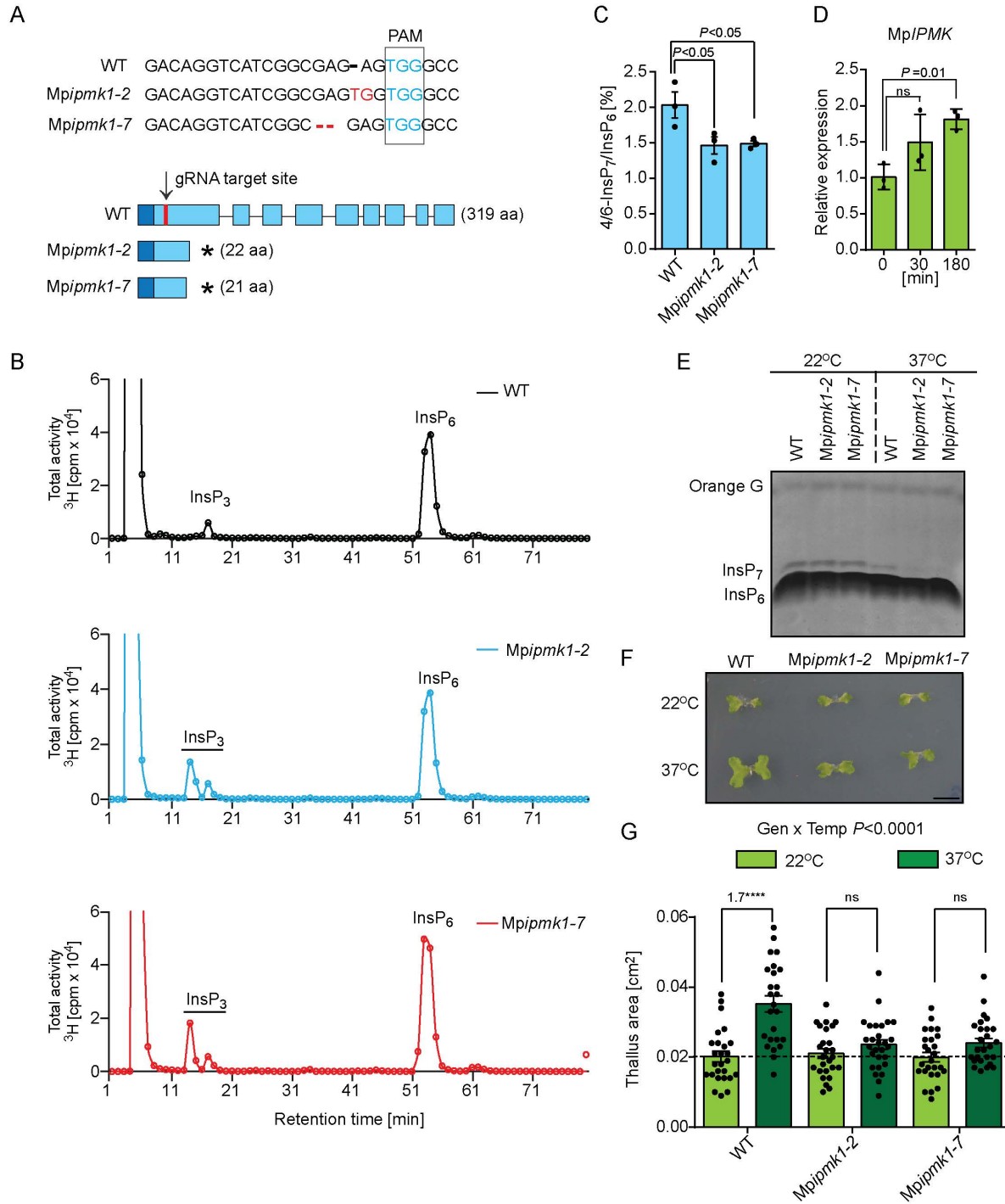

**Fig 6. MpIPMK activity is essential for maintaining the major pool of InsP$_7$ during heat stress and it regulates heat stress acclimation in *M. polymorpha*. A.** Generation of Mp*ipmk* knockout lines. Schematic representation of the two independent Mp*ipmk* knockout lines obtained by CRISPR-Cas9 gene editing technology. Mp*ipmk1-2* has one insertion and one substitution of nucleotides preceding PAM sequence while Mp*ipmk1-7* has deletion of two nucleotides preceding PAM sequence. The mutations resulted in a premature stop codon. **B.** IPMK activity is not critical for maintaining InsP$_6$ level in *M. polymorpha* thallus. SAX-HPLC analysis of [$^3$H]-*myo*-inositol-labelled wild-type and Mp*ipmk* knockout plants. Neutralized extracts were resolved by SAX-HPLC and fractions collected each minute for subsequent determination of radioactivity as indicated. **C.** MpIPMK-deficient plants are compromised in 4/6-InsP$_7$ production. CE-MS analyses of 4/6-InsP$_7$ level in 14-day-old thalli of wild-type, Mp*ipmk1-2* and Mp*ipmk 1-7* knockout

lines. 4/6-InsP$_7$ is presented in percentage to InsP$_6$. Data are means ± SEM (n = 3, biological replicates). Significant difference is determined by one-way ANOVA followed by Dunnett's test. **D.** Quantitative RT-PCR (qRT-PCR) analysis of Mp*IPMK* expression in wild-type thallus exposed to heat stress at 37°C for different time intervals. Mp*ACT7* was used for normalization. Statistical significance is determined by one-way analysis of variance (ANOVA) followed by Dunnett's test. **E.** MpIPMK activity is critical to maintain the major pool of InsP$_7$ production during heat stress. Inositol phosphates were extracted using TiO$_2$ beads from 14-day-old thalli of wild-type and Mp*ipmk* knockout lines grown at 22°C and exposed to heat stress at 37°C for 3 h. InsPs were separated by 33% PAGE and visualized by toluidine blue stain. **F-G.** MpIPMK activity is critical for inducing heat stress response. Photograph showing representative thallus of the indicated genotypes grown at 22°C (top panel) and grown at 37°C for 8 h in growth chamber with subsequent recovery at 22°C (bottom panel) for 25 days (F). Quantification of thallus area as a measure of thermomorphogenic response (G). 5-day-old gemmalings of wild-type and Mp*ipmk* knockout lines were subjected to heat stress in plant chamber maintained at 37°C for 8 h followed by subsequent recovery at 22°C for 13 days. Values are means ±SEM (n = 3, biological replicates) and significant difference is determined by one-way analysis of variance (ANOVA) followed by Tukey's test (****$P < 0.0001$). Number on the bar depicts the fold changes in the thallus area upon heat stress compared to control condition.

that the cellular levels of InsP$_7$ were severely affected in the independent Mp*ipmk* mutant plants, compared to the wild-type line during elevated heat stress (Fig 6E).

To investigate the consequences of loss of MpIPMK activity in thermomorphogenesis, we monitored the response of wild-type and the two independent Mp*ipmk* mutant plants after exposing them to the elevated temperature, 37°C for 8 h. In line with previous reports [82,83], heat treatment resulted in increased thallus area of the wild-type plants (Fig 6F and 6G). In contrast, the Mp*ipmk* mutants displayed severe reduction in thallus area compared to the wild-type plants (Fig 6F and 6G). Taken together, these data show that MpIPMK critically contributes to plant resilience to heat stress.

### *M. polymorpha* IPMK rescues the altered heat stress tolerance of *Arabidopsis* IPK2-defective plants

To investigate the functional conservation of IPK2 proteins between *M. polymorpha* and *Arabidopsis*, we generated the *atipk2βatipk2α*$^{kd}$ transgenic lines expressing Mp*IPMK* under the control of a constitutive 35S promoter (S10A and S10B Fig) and performed shoot thermomorphogenesis assays. Remarkably, heterologous expression of Mp*IPMK* rescued the attenuated heat-induced hypocotyl elongation of the *atipk2βatipk2α*$^{kd}$ plants (Fig 7A and 7B). Similarly, the expression of Mp*IPMK* largely rescued the compromised survival rate of *atipk2β atipk2α*$^{kd}$ plants during heat stress (Fig 7C). Altogether, our findings highlight the functional conservation of IPK2 proteins between liverworts and angiosperms.

### IPK2 promotes the transcriptional activity of HSF through both catalytic-dependent and catalytic-independent mechanisms

To gain possible mechanistic insights about IPK2-controlled heat stress acclimation, we first tested whether genes encoding members of HSP families, PIF, and genes involved in auxin signalling pathway that play role in heat stress acclimation [76,84–88], are differentially expressed in the *Arabidopsis AtIPK2* knockdown lines. Our analyses suggest that the expression of different *HSP*s including *HSP70*, *HSP22*, *HSP17.6*, and *HSP18.1* is largely compromised in the *atipk2βatipk2α*$^{kd}$ lines when compared to Col-0 plants during heat stress (Fig 8A). Furthermore, the *atipk2βatipk2α*$^{kd}$ lines showed deregulated expression of heat stress responsive genes involved in auxin signaling pathways, e.g., *IAA19* and *YUC8* (S10C Fig). Considering that the *atipk2βatipk2α*$^{kd}$ lines suffered with the compromised expression of genes that are regulated by different heat shock transcription factors (HSF-TFs) [87,89,90] and that IPMK/IPK2 is already implicated in transcriptional regulation in various eukaryotes [21,91–94], we asked whether IPK2 controls HSF activity *in planta*. To this end, we performed transient transcription assay using dual-luciferase (LUC) reporter plasmid in *N. benthamiana* leaves. The plasmid encodes a *Renilla* luciferase gene driven by the constitutive 35S promoter, and a firefly *LUC* gene driven by the *AtHSP18.1* promoter. Our analyses indicate that the presence of AtIPK2 significantly enhanced the transcriptional activity of AtHSFA1b (Fig 8B). In contrast, the catalytic dead variant of AtIPK2α did not augment the activity of AtHSFA1b (Fig 8B). Collectively, these results suggest that the catalytic activity of *Arabidopsis* IPK2α enhances the transcriptional activity of AtHSFA1b, critical for plant adaptation to heat stress.

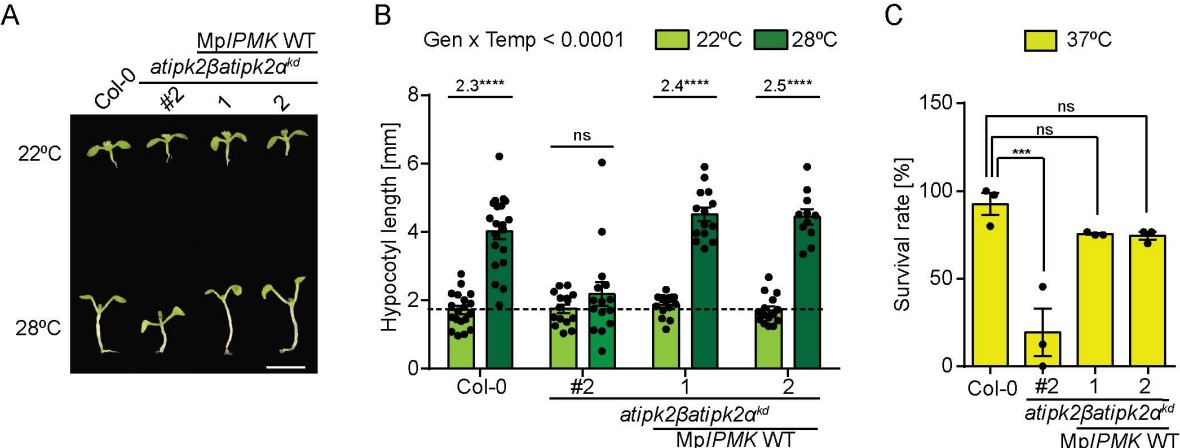

**Fig 7. *Arabidopsis* and *M. polymorpha* share a functional IPK2 homolog. A.** Heterologous expression of Mp*IPMK* rescues attenuated hypocotyl elongation of the *atipk2βatipk2α^kd* lines during heat stress. Representative photograph of high temperature-induced hypocotyl elongation phenotype of the indicated genotypes grown at 22ºC and 28ºC. 7-day-old seedlings were exposed to 28ºC for 5 days. Control plates were maintained at 22ºC. Images were captured after 5 days of heat stress. Scale bar = 6 mm. **B.** Quantification of elongated hypocotyls. Data are means ± SEM (n ≥ 11, biological replicates). Statistical significance determined by two-way analysis of variance (ANOVA) followed by Tukey's test (****$P < 0.0001$). Numbers on the bar represents the fold change in hypocotyl length of the designated genotypes. Dashed line depicts the comparable hypocotyl length of the studied genotypes. **C.** Heterologous expression of Mp*IPMK* enhances the basal thermotolerance of *atipk2βatipk2α^kd* lines during heat stress. Quantification of survival rate of the designated genotypes after heat stress at 37ºC. Surviving seedlings maintain green leaves and show emerged new leaves. Values are means ± SEM (n = 3, biological replicates) with each data point indicated and significant difference is determined by one-way analysis of variance (ANOVA) followed by Dunnett's test (***$P < 0.001$).

Given that the catalytic-independent activity of IPMK/IPK2 has already been implicated in transcriptional regulation across various eukaryotes [21,91–95], we investigated whether IPMK directly regulates HSF activity. To elucidate the role of IPMK-type proteins in transcriptional regulation, we first assessed whether these proteins physically associate with heat shock factors (HSFs) and subsequently influence their DNA-binding activity. Using yeast two hybrid (Y2H) assay and Bimolecular Fluorescence Complementation (BiFC) assay, we show that AtIPK2β physically interacts with AtHSAF1b (Fig 9A and 9B). Next, to investigate the consequence of this physical interaction, we performed an electrophoretic mobility shift assay (EMSA). The *in vitro* DNA binding study was performed using a short double nucleotide fragment containing the heat shock promoter element (HSE) of *AtHSP18.1*. In agreement with the previous report [96], we found that AtHSFA1b binds specifically to the HSE element (Fig 9C, 9D and 9E). Notably, when AtIPK2β was incubated with AtHSFA1b, a strong shift of the DNA occurred in a dose-dependent manner compared to incubation with AtHSFA1b alone, suggesting that AtIPK2β modulates the DNA-binding activity of HSFA1b (Fig 9F). Notably, the DNA-binding activity of HSFA1b was not influenced by the presence of MBP (Fig 9G). Importantly, AtIPK2β and MBP alone were unable to bind HSE element, suggesting that the enhancement in promoter element binding activity of HSFs in presence of AtIPK2β is not due to direct binding of AtIPK2β proteins with the promoter element (S11A and S11B Fig). Notably, incubation of AtHSFA1b with Ins(1,4,5)P$_3$, InsP$_6$, and 4-InsP$_7$ didn't affect the DNA binding activity of AtHSFA1b (S11C Fig). Similar to Arabidopsis IPK2 proteins, we found that Marchantia IPMK physically interacts with MpHSFB1 (S12A and 12B Fig). Our EMSA analyses revealed that similar to AtHSFA1b, MpHSFB1 shows specific binding to HSE element (Fig 9H-9J). Intriguingly, MpIPMK also potentiates the DNA binding activity of MpHSFB1 in a dose-dependent manner (Fig 9K), whereas MBP fails to elicit a similar effect (Fig 9L). Moreover, MpIPMK alone did not show binding to the promoter element (S12C Fig). Additionally, the tested InsP species could not influence the DNA binding activity of MpHSFB1 *in vitro* (S12D Fig). Collectively, these findings highlight the crucial catalytic-independent function of IPK2 in controlling the DNA-binding activity of HSF-type transcription factors.

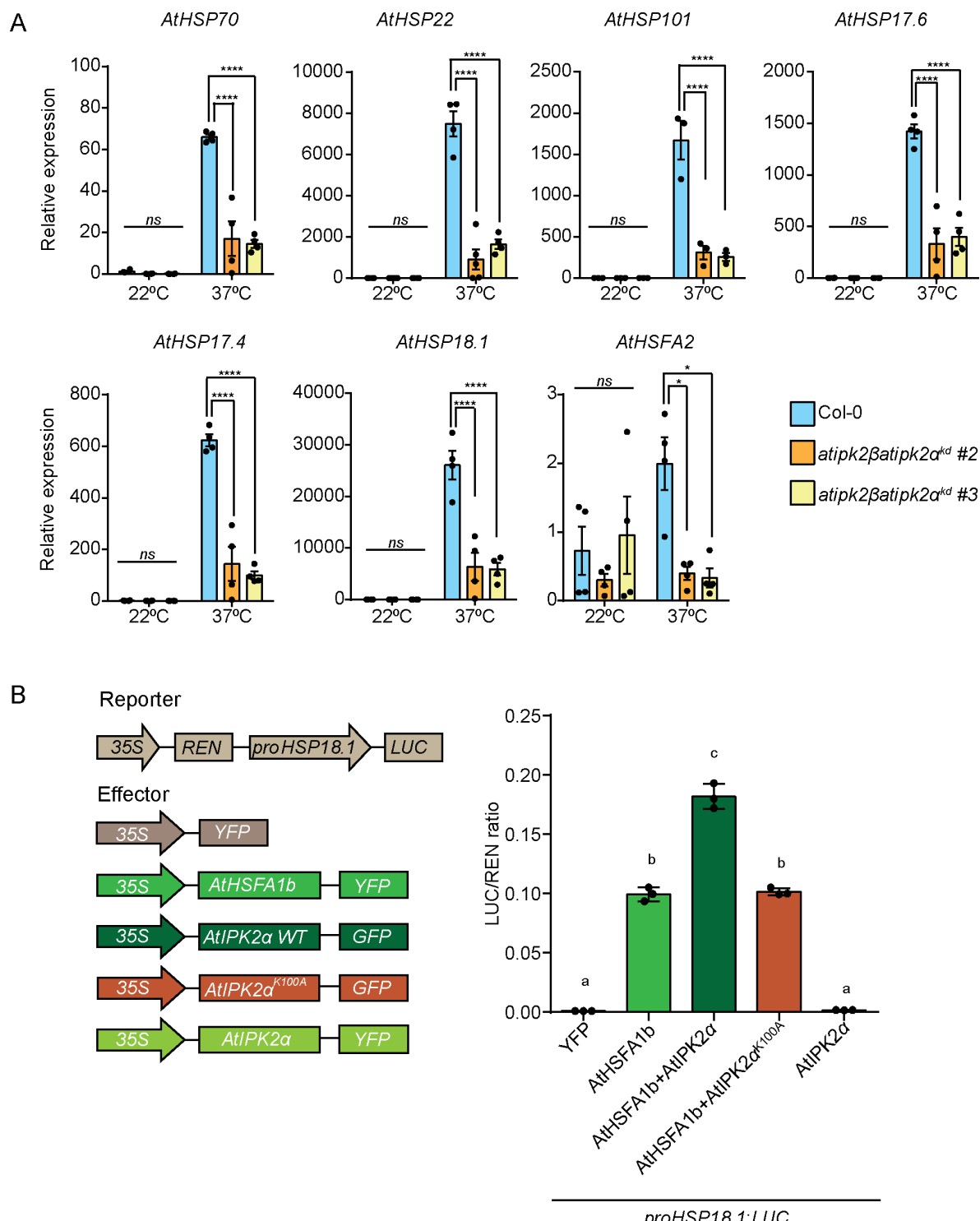

**Fig 8. AtIPK2α modulates the transcriptional activity of HSFs. A.** Quantitative RT-PCR (qRT-PCR) analysis of different *HSP*s in Col-0 and *atip-k2βatipk2αkd* lines after heat shock. 14-day-old seedlings were exposed to 37°C for 3h and were harvested for qRT-PCR analysis. *PP2AA3* was used as a reference gene. Values are means±SEM (n=3, biological replicates). Statistical significance is determined by two-way ANOVA followed by Tukey's test (*P<0.05, ****P<0.0001). **B.** AtIPK2α potentiates transcription activity of heat shock transcription factor, AtHSFA1b *in planta*. Schematic diagrams of

luciferase reporter and effector constructs used in transient transactivation assays in *Nicotiana benthamiana* leaves (left panel). Statistical analysis of the expression of *pHSP18.1-LUC* in presence of AtHSFA1b and AtIPK2α (right panel). YFP served as control. Values are means ±SEM (n = 3, biological replicates). Different letters indicate significance in one-way analysis of variance (ANOVA) followed by Tukey's test (a and b, $P < 0.0001$; b and c, $P < 0.0001$; a and c, $P < 0.0001$).

## Discussion

PP-InsPs serve as cellular messengers that control a wide-range of physiological processes in eukaryotes. Using CE-MS and nuclear magnetic resonance (NMR) spectroscopy, several PP-InsP species have been identified in *Arabidopsis* extracts [30,33,34,56]. In this study, we report that 4/6-InsP$_7$ is the predominant PP-InsP species present in the studied land plant tissue extracts. In future studies, we will monitor the distribution of PP-InsP species in different plant tissues, different stages of plant development and do so across diverse plant species. Currently, we are unable to determine whether it is 4-InsP$_7$ or 6-InsP$_7$, i.e., which enantiomer is produced by IPK2 preferentially. Although these are interesting questions to be addressed, the major roadblocks to answer them are: i) conventional CE-MS cannot differentiate between enantiomers, ii) although NMR spectroscopy in presence of a chiral solvating agent could be employed to illustrate the enantiomeric identity of 4/6-InsP$_7$ as described previously to identify the product of a bacterial effector protein [97], 4/6-InsP$_7$ purified from *Arabidopsis* seedling extracts is not adequate for such NMR analysis. Structural elucidation of a plant-derived InsP$_7$ species using NMR was reported for *atmrp5* mutant seeds [30], that are defective in vacuolar InsP$_6$ loading, consequently, cyto-/nucleoplasmic levels of InsP$_6$-derived PP-InsP species are augmented in the AtMRP5-deficient plants [36,98,99]. Further investigations are required to optimize purification of 4/6-InsP$_7$ from plant extracts for the structural determination of plant-derived PP-InsP species by NMR.

In search for putative kinases responsible for 4/6-InsP$_7$ production in plants, we found AtIPK2α as one of the primary candidates through a structure-based homology screen. Previous studies had reported that similar to yeast Ipk2, AtIPK2 proteins phosphorylate Ins(1,4,5)P$_3$ at 6-OH and 3-OH positions, respectively to generate Ins(1,3,4,5,6)P$_5$ *in vitro* [21,22,51,53]. However, contribution of AtIPK2 homologs in inositol phosphate homeostasis *in planta* remained largely obscure, mostly due to their redundancy. Additionally, the role of AtIPK2 as a PP-InsP synthase was not established previously. We show that AtIPK2α and AtIPK2β bind to InsP$_6$ with strong affinity, suggesting that InsP$_6$ could be a substrate of AtIPK2. Our kinetics analysis revealed that *Km* ATP for InsP$_6$ of AtIPK2 is somewhat comparable to that of mammalian IP6K and Arabidopsis ITPK1, reinstating that InsP$_6$ could be a possible physiological substrate of AtIPK2 enzymes. Consistent with the role of AtIPK2 as a PP-InsP synthase, we show that *Arabidopsis* IPK2 proteins can phosphorylate InsP$_6$ to generate 4/6-InsP$_7$ *in vitro*.

Do AtIPK2α and AtIPK2β play a role in InsP and PP-InsP metabolism *in planta*? To explore this, we analyzed the individual *atipk2α-1* and *atipk2β-1* knockout plants and observed a similar profile when compared to Col-0 seedlings. Notably, the *atipk2α-1* plants exhibited elevated levels of an InsP$_5$ isomer while InsP$_6$ and PP-InsP isomers remained unchanged. To mitigate possible functional redundancy between AtIPK2α and AtIPK2β, we generated *Arabidopsis* transgenic lines with reduced transcript of At*IPK2α* in the *atipk2β-1* knockout plant background. Our CE-MS analyses of knockdown lines revealed that certain InsP$_4$ and InsP$_5$ isomers are deregulated in the *atipk2βatipk2α$^{kd}$* plants, yet InsP$_6$ levels remained unaffected in the AtIPK2-deficient plants. Collectively, these data suggest that while AtIPK2 regulates lower inositol phosphate metabolism, and that deregulation of InsP$_{4-5}$ metabolism does not affect the global pool of InsP$_6$ in seedlings. This differs from yeast and mammalian systems where IPK2/IPMK activity is critical for maintaining cellular levels of InsP$_6$ [21,22,49]. The consequence of altered InsP$_4$ and InsP$_5$ isomers in the *atipk2βatipk2α$^{kd}$* seedlings is yet to be understood. In agreement with previous reports [36,50,62,63], we found that seed phytate level is controlled by both AtIPK2α and AtIPK2β highlighting specialized functions of the AtIPK2 members in different plant parts. CE-MS analyses revealed a significant decrease in 4/6-InsP$_7$ in the *Arabidopsis* IPK2-deficient

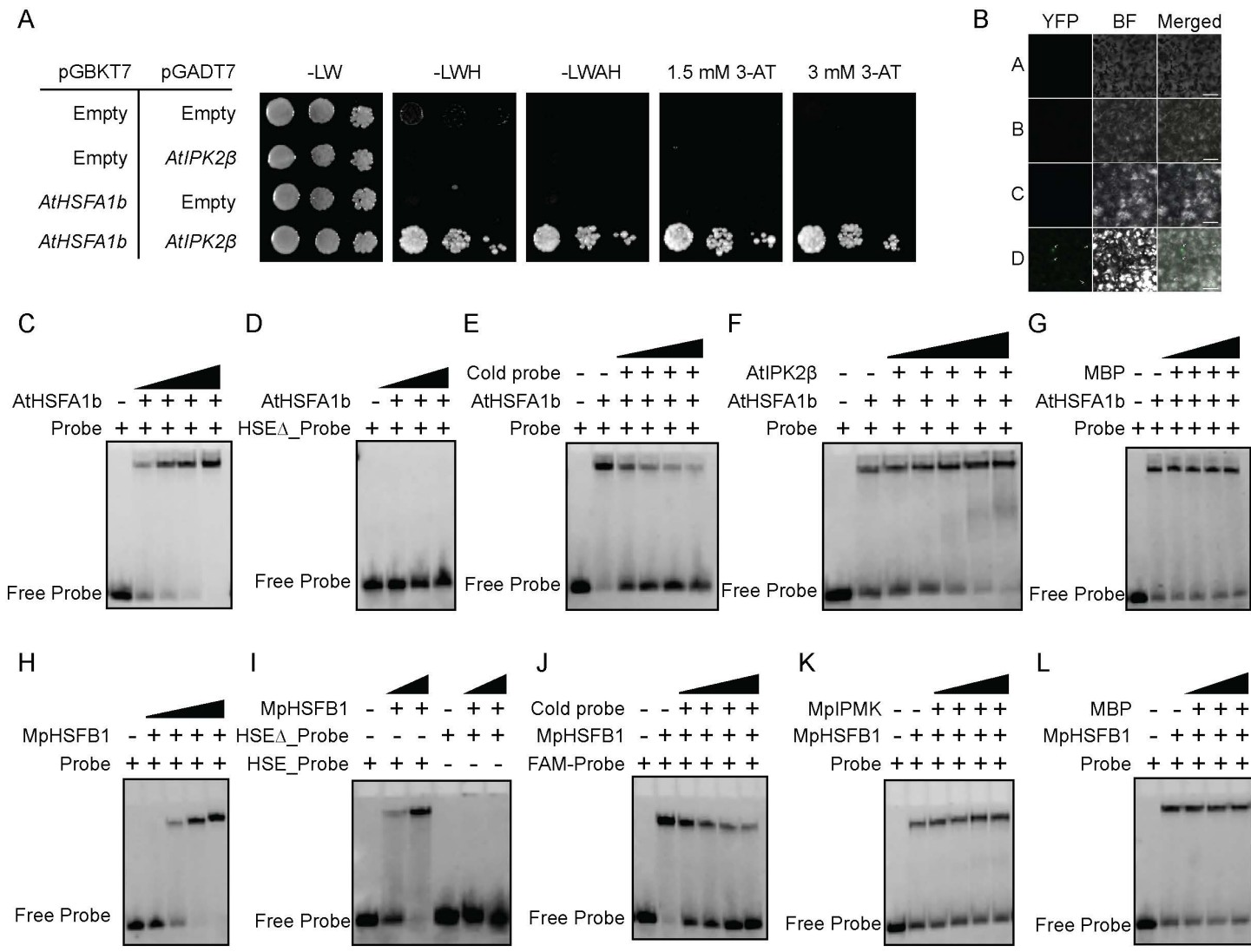

**Fig 9. IPK2-type proteins interact physically with HSFs and enhance the DNA-binding activity of HSF *in vitro*. A.** AtIPK2β exhibits physical interaction with AtHSFA1b *in vivo*. AH109 yeast strain carrying the pGADT7-AtIPK2β and pGBKT7-AtHSFA1b plasmids were spotted on selective media lacking leucine (Leu, L), tryptophan (Trp, W), histidine (His, H), adenine (Ade, A) and indicated amount of 3-AT. **B.** BiFC experiment showing physical interaction of IPK2β and AtHSFA1b in *N. benthamiana* leaves. Empty vectors (pMDC_nVenus/ pMDC_CFP) and different combinations of empty vectors and cloned constructs of *AtHSFA1b* and *AtIPK2β* served as negative control. A-D denotes different *A. tumefaciens* transformants harbouring combination of vectors co-infiltrated in *N. benthamiana* leaves where A = cCFP empty + nVENUS empty, B = cCFP_AtIPK2β + nVENUS_empty, C = cCFP_empty + nVE-NUS_AtHSFA1b, D = cCFP_AtIPK2β + nVENUS_AtHSFA1b. **C.** An electrophoretic mobility assay (EMSA) showing AtHSFA1b binds to FAM-labelled *HSP18.1* promoter element having canonical HSE element (GAAnnTTC) in a dose-dependent manner. 250 nM of the FAM-labelled probe was incubated with recombinant MBP-AtHSFA1b (25, 50, 100 and 200 nM) for 15 mins on ice and resolved on 6% of native PAGE. **D.** EMSA showing AtHSFA1b doesn't bind to probe lacking HSE element. 250 nM of the mutant probe (HSEΔ) was incubated with recombinant MBP-AtHSFA1b (50, 100 and 200 nM) for 15 mins on ice and resolved on 6% of native PAGE. **E.** EMSA showing competition between the FAM-labelled and unlabelled probe for AtHSFA1b binding. 250 nM of both FAM-labelled and different concentration of unlabelled probe (50, 100, 250 nM, 500 nM), was incubated with 50 nM of AtHSFA1b for 15 mins on ice and resolved using 6% native PAGE. **F.** EMSA showing AtIPK2β enhances DNA binding activity of AtHSFA1b in a dose-dependent manner. Recombinant MBP-AtIPK2β (25, 50, 100, 200 and 500 nM) was incubated with 50 nM of recombinant MBP-AtHSFA1b for 30 mins followed by post incubated with 250 nM of FAM-labelled probe for 15 mins on ice. The complexes were resolved using 6% PAGE. **G.** EMSA showing negative control MBP cannot enhance DNA-binding activity of MpHSFB1 in a dose-dependent manner. Recombinant His8-MBP (25, 50, 100 and 250 nM) was incubated with recombinant MBP-AtHSFA1b (50 nM) for 30 mins on ice followed by post incubation with 250 nM of FAM-labelled probe for 15 mins on ice. The complexes were resolved using 6% native PAGE. **H.** An electrophoretic mobility assay (EMSA) showing MpHSFB1 binds to FAM-labelled *Heat shock element (HSE)* in a dose-dependent manner. 250 nM of the FAM-labelled probe was incubated with recombinant MBP-MpHSFB1 (25, 50, 100 and 250 nM) for 15 mins on ice and resolved on 6% of native PAGE. **I.** EMSA showing MpHSFB1 doesn't bind to probe lacking HSE element. 250 nM of the mutant

probe (HSEΔ) was incubated with recombinant MBP-MpHSFB1 (50 and 100 nM). **J.** EMSA showing competition between the FAM-labelled and unlabelled probe for MpHSFB1 binding. 250 nM of both FAM-labelled and unlabelled probe (50, 100, 250, 500 nM) was incubated with 50 nM of MpHSFB1 for 15 mins on ice and resolved using 6% native PAGE. **K.** EMSA showing MpIPMK enhances DNA binding activity of MpHSFB1 in a dose-dependent manner. Recombinant MBP-MpIPMK (50, 100, 250 and 500 nM) was incubated with 50 nM of recombinant MBP-MpHSFB1 for 30 mins followed by post incubated with 250 nM of FAM-labelled probe for 15 mins on ice. The complexes were resolved using 6% PAGE. **L.** EMSA showing negative control MBP cannot enhance DNA-binding activity of MpHSFB1. Recombinant His8-MBP (50, 100, 250 and 500 nM) was incubated with recombinant MBP-MpHSFB1 (50 nM) for 30 mins on ice followed by post incubation with 250 nM of FAM-labelled probe for 15 mins on ice. The complexes were resolved using 6% native PAGE.

seedlings, while the 5-InsP$_7$ and 1/3-InsP$_7$ isomers remained unaffected. This suggests that IPK2 specifically regulates the cellular level of 4/6-InsP$_7$. It remains to be determined whether Arabidopsis IPK2 can generate a PP-InsP isomer using monophosphate-containing InsPs other than InsP$_6$ as substrates. Future work awaits to clarify whether IPK2 can produce 4/6-InsP$_7$ in an InsP$_6$-independent manner using InsP$_{3/4/5}$ as substrates *in planta*. Since the *atipk2βatipk2α* double knockout plants are embryonic lethal and that the *atipk2βatipk2α$^{kd}$* plants may retain residual AtIPK2 activity, it is yet to be explored whether the *Arabidopsis* genome encodes protein(s) other than AtIPK2 homologs that contribute to the remaining pool of 4/6-InsP$_7$ present in the knockdown lines. In agreement with the role of AtIPK2 in heat stress acclimation, the *atipk2βatipk2α$^{kd}$* plants displayed compromised shoot thermomorphogenesis. Furthermore, basal thermotolerance of the knockdown lines is severely affected. Notably, the altered heat stress acclimation of the IPK2-deficient plants could be reversed by ectopic expression of *AtIPK2β* under the control of a constitutive promoter.

Our parallel investigation using *M. polymorpha* allowed us to conclude that IPK2-dependent 4/6-InsP$_7$ production is an ancestral function, and that MpIPMK contributes to 4/6-InsP$_7$ production *in planta*. Future work awaits to clarify whether the *M. polymorpha* genome encodes proteins other than MpIPMK to control 4/6-InsP$_7$ synthesis. As discussed above, *M. polymorpha* might possess both 4-InsP$_7$ and 6-InsP$_7$ in a different ratio compared to *Arabidopsis* yet, it is also conceivable and likely that IPMK/IPK2 catalyze the synthesis of only one of them with high enantioselectivity. These are unresolved questions that require further investigation. Despite the accumulation of certain InsP$_3$, InsP$_4$ and InsP$_5$ species in both *Arabidopsis* and *M. polymorpha* IPK2/IPMK-deficient lines, the unaltered InsP$_6$ level suggests that the archetypal activity of IPK2 in producing InsP$_6$ (also known as phytic acid) in yeast is not conserved in the vegetative tissues of land plants. Our data also indicates that plant IPK2 evolved a distinct PP-InsP synthase activity during their divergence from the fungal lineage.

The ectopic expression of Mp*IPMK* fully rescued the altered thermomorphogenic responses of the *atipk2βatipk2α$^{kd}$* plants, suggesting that *Arabidopsis* and *M. polymorpha* share functional IPK2 homologs. Moreover, our study sheds light on mechanistic insights into the regulation of heat stress acclimation by IPK2. Specifically, AtIPK2α augments DNA-binding and transcription activity of AtHSFA1b. In contrast, the catalytic dead variant of AtIPK2α does not potentiate the transcription activity of AtHSFA1b *in planta*, highlighting role AtIPK2-derived InsP and PP-InsP species in heat stress acclimation. Future work awaits to clarify which of the AtIPK2-dependent inositol phosphates regulate the transcription activity of AtHSFA1b. Although the reduction of 4/6-InsP$_7$ during heat stress points towards a possible function of this PP-InsP isomer in heat stress acclimation, future studies need to address whether reduction of 4/6-InsP$_7$ is specific to certain duration of heat exposure and the type of heat exposure (i.e., 28ºC vs. 37ºC). Notably, our findings also suggest that IPK2 can augment DNA binding activity of HSFs *in vitro*, highlighting the importance of catalytic-independent activity of IPK2 in heat stress acclimation. Further investigation is required to dissect the contribution of catalytic vs. non-catalytic activity of IPK2 proteins during heat stress.

In conclusion, our study offers a mechanistic framework to understand the conserved role of IPK2-derived InsP and PP-InsP species in regulating various cellular processes, and how this contributes to land plant evolution.

## Methods

### Phylogenetic analysis

The phylogenetic tree was constructed as described previously [35]. BLAST search analyses (https://blast.ncbi.nlm.nih.gov/Blast.cgi?PAGE=Proteins) were performed using the full-length sequence of AtIPK2α retrieved from *A. thaliana* genome database (https://www.arabidopsis.org/). The retrieved sequences were filtered to include only those sequences with a percent identity of more than 30%, query cover of more than 35%, E-value of less than $10^{-5}$ and a bit score of more than 80. Sequences from every species were screened for the presence of gene isoforms. The GUIDANCE2 server (http://guidance.tau.ac.il/) was used to align the sequences using the version of MAFFT available on the server. The GUIDANCE2 algorithm was used to estimate the reliability of alignment columns. Unreliable columns having a confidence score lesser than 0.93 were removed from the multiple sequence alignment (GUIDANCE Server - a web server for assessing alignment confidence score (tau.ac.il). Phylogenetic analysis was conducted in MEGA11 [100]. The best maximum likelihood model for the given alignment was estimated using the default settings and this model was used to estimate a maximum likelihood tree. Alignment of complete protein sequences were done using ClustalW and phylogenetic tree was constructed with the Maximum Likelihood method, using the Dayhoff's Model and a bootstrap test of 1000 replicates. Values less than 50% are not displayed on the tree and branch lengths are given in terms of the expected number of amino acid substitutions per site.

### Plant materials and growth conditions

Seeds of *Arabidopsis thaliana* T-DNA insertional mutant of *ipk2β-1* (SALK_104995) was genotyped for homozygous T-DNA insertions using primer S1 Table. Surface sterilization of seeds was performed by incubating the seeds in solution containing 0.05% SDS in 70% ethanol for 15 min. Sterilized seeds were sown onto the solidified half-strength Murashige and Skoog (MS) media containing 0.8% agar (w/v). After stratification for 3 days at 4 °C, plates were transferred in a Percival plant chamber under conditions of 16 h light and 8 h dark at 22 °C with light intensity 100 µmol/m$^2$/s. The germinated seedlings were transferred onto soil (perlite and soilrite in the ratio of 1:2) and maintained in growth room. The growth room condition was maintained at 22°C with 70% RH and long-day (LD) conditions (16 h:8 h; light: dark cycle) with light intensity 100 µmol/m$^2$/s.

   *Marchantia polymorpha* accession Takaragaike-1 (Tak-1, male accession) and Takaragaike-2 (Tak-2, female accession) were used as wild-type plants. *M. polymorpha* lines were propagated asexually by gemma cultured on half-strength Gamborg's B5 medium with 1% phytagel under long-day (LD) conditions (16 h:8 h; light: dark cycle) with light intensity 50–60 µmol/m$^2$/s in a Percival growth chamber at 22ºC.

   Mature sporophylls of *Nephrolepis sp.* and *Dryopteris sp.* were collected from IISc campus.

### Molecular cloning

Cloning of *AtIPK2* and Mp*IPMK* in pET28b-His$_8$-MBP bacterial and yeast expression vector.

   Full-length coding sequence of AtIPK2α, AtIPK2β and MpIPMK were amplified using cDNA prepared from total RNA extracts of Col-0 and Tak-1 plants, respectively with primers listed in S1 Table. The amplified products were cloned at the EcoRV site in pBLUESCRIPT via blunt end cloning followed by directional subcloning into pET28b between the BamHI and Not1 sites. PCR mutagenesis was used to generate a point mutation in the conserved PXXXDXKXG motif of *AtIPK2α, AtIPK2β* and Mp*IPMK.* Primer used to generate site directed mutagens are enlisted in S1 Table. For cloning Mp*IPMK* wild-type and catalytic dead variants in yeast expression vector pCA45, the amplified product of Mp*IPMK* coding DNA sequence (CDS) from Tak-1 plants was subcloned with pBLUESCRIPT followed by directional cloning in pCA45 vector pre-digested with the BamH1 and Not1.

## Protein expression and purification

To purify recombinant AtIPK2α and AtIPK2β proteins, *E. coli* BL21 (RIL) strains were transformed with the pET28b-His$_8$-MBP-AtIPK2α and pET28b-His$_8$-MBP-AtIPK2β vectors. A single colony of transformants carrying the respective constructs was used to inoculate in terrific broth (TB medium). After induction with 0.5 mM IPTG, the culture was allowed to incubate further for 3 days at 12ºC. Cells were harvested at 6000 rpm for 10 min at 4ºC and pellet was washed with lysis buffer (300 mM NaCl, 25 mM Na$_2$HPO$_4$, pH 7.5). The pellets were resuspended in lysis buffer containing 5 mM β-ME and 1 mM PMSF, followed by cell lysis using sonication at 10 pulse of 30 sec 'ON' and 10 sec 'OFF'. After sonication, the lysates were centrifuged at high speed 18000 rpm for 45 min at 4ºC. Meanwhile Ni-NTA resin (Qiagen) was prepared by washing with ultra-pure water and the equilibrated with lysis buffer twice. After centrifugation, the cleared protein supernatant was incubated with prewashed Ni-NTA beads for 6 h on rotor at 4ºC. Next, the beads were washed thrice with wash buffer (lysis buffer, 5 mM β-ME, 10 mM imidazole) at 4000 rpm for 5 min at 4ºC. For elution, the beads were incubated with 250 µL elution buffer (lysis buffer containing 5 mM β-ME, 250 mM imidazole) at 4ºC for 5 min on a rotor and centrifuged at 4000 rpm for 5 mins. Three such elutions were collected. Aliquots of elution and different concentration of BSA standards were heated at 95ºC after adding 1X SDS loading dye and loaded on 12% SDS-PAGE. Protein was visualized by coomassie staining and quantification of the band intensity was done using ImageJ. The mutant variants of AtIPK2α and AtIPK2β purification were performed as mentioned above. Recombinant MpIPMK protein and its catalytic dead variants were purified using Phosphate Buffer Saline (PBS) buffer [137 mM NaCl, 2.7 mM KCl, 10 mM Na$_2$HPO$_4$, 1.8 mM KH$_2$PO$_4$ (pH 7.4)].

## Isothermal titration calorimetry (ITC)

For ITC, tag-less AtIPK2 proteins were generated by subjecting the purified His$_8$-MBP-TEV-AtIPK2α/β recombinant proteins to TEV protease dialysed in 50 mM Tris Cl pH 8, 0.5 mM EDTA followed by binding with Ni-NTA and amylose resins. All ITC experiments were performed at 25 °C using a MicroCal PEAQ-ITC system (Malvern Panalytical) equipped with a 280 µL sample cell and a 40 µL injection syringe. All proteins were dialyzed against buffer (50 mM Tris Cl pH 8.0 and 0.5 mM EDTA); Ins(1,4,5)P$_3$ and InsP$_6$ ligands were diluted in same buffer prior to all measurements. A typical titration consisted of 19 injections, the protein concentrations in the syringe and in the cell are provided in the respective figure legend. Data were analyzed using the MicroCal PEAQ-ITC analysis software (v1.21). InsP$_3$ and InsP$_6$ were obtained from SiChem GmbH.

## *In vitro* kinase assay

The InsP$_6$ kinase assays were performed by incubating 0.16 µg µL$^{-1}$ of recombinant AtIPK2α and AtIPK2β and their catalytic dead variants in a 30 µL of reaction volume containing kinase buffer [5 mM MgCl$_2$, 20 mM HEPES (pH 7.5), 1 mM DTT], 12.5 mM ATP, and 10 nmol InsP$_6$ at 37ºC for 12 h. MBP protein was used as negative control. The *in vitro* kinase assay of MpIPMK and its catalytic dead variants was performed using the above-mentioned reaction condition. The reaction products were resolved using PAGE and visualized by toluidine blue staining [64].

For *K*m and *V*max calculations of ATP, 0.06 µg µL$^{-1}$ of AtIPK2β and AtIPK2α was incubated with InsP$_6$ and assayed with 2–12 mM ATP for 10 h. For time course experiment AtIPK2β was incubated with 12.5 mM ATP, and 10 nmol InsP$_6$ at 37ºC for different time intervals. The reaction products were analysed using CE-MS.

## Enrichment of inositol phosphate using titanium dioxide bead

Inositol phosphate pull down was performed as described previously [71]. All steps were carried on ice. TiO$_2$ beads were weighed to 10–12 mg for each sample and washed twice in water and once in 1 M perchloric acid (PA). Liquid N$_2$-frozen plant material (approx. ~100 mg) was homogenized using a pestle and immediately resuspended in 800 µL ice-cold PA. Samples were kept on ice for 5 min with short intermediate vortexing and then centrifuged for 5 min at 20000 g at 4ºC. The

supernatants were transferred into fresh 1.5 mL tubes and centrifuged again for 5 min at 20000 g. The supernatants were resuspended in the prewashed $TiO_2$ beads and incubated at 4ºC for 30 min. After incubation, the beads were pelleted by centrifuging at 8000 g for 1 min and washed twice in PA. The supernatants were discarded. To elute inositol polyphosphates, beads were resuspended in 200 μL of 10% ammonium hydroxide and then incubated for 5 min at room temperature. After centrifuging at 2600 g, the supernatants were transferred into fresh tubes. The elution process was repeated, and the second supernatants were pooled as well. Eluted samples were vacuum evaporated at room temperature. InsPs were resuspended in 40 μL ultra-pure water for the following CE-MS analysis.

## CE-MS analysis

The analysis was performed on a CE-QQQ system (Agilent 7100 CE-with Agilent 6495C Triple Quadrupole and Agilent Jet Stream electrospray ionization source, adopting an Agilent CE-ESI-MS interface). An isocratic Agilent 1200 LC pump was used to deliver the sheath liquid (50% isopropanol in water) with a final splitting flow speed of 10 μL/min via a splitter. All separation was performed via a bare fused silica capillary with a length of 100 cm (50 μm internal diameter and 365 μm outer diameter). 40 mM ammonium acetate titrated with ammonium hydroxide to pH 9.0 was used as background electrolyte (BGE). Between runs of each sample, the capillary was flushed with BGE for 400s. Samples were injected by applying 100 mbar pressure for 15s (30 nL). The MS source parameters were as follows: gas temperature was 150˚C, gas flow was 11 L/min, nebulizer pressure was 8 psi, sheath gas temperature was 175˚C and with a flow at 8 L/min, the capillary voltage was -2000V, the nozzle voltage was 2000V. Negative high-pressure RF and negative low-pressure RF were 70 V and 40 V, respectively. Multiple reaction monitoring (MRM) transitions were setting as shown in S6 Table.

Internal standard (IS) stock solution of 8 μM [$^{13}C_6$] 2-OH $InsP_5$, 40 μM [$^{13}C_6$] $InsP_6$, 2 μM [$^{13}C_6$] 1-$InsP_7$, 2 μM [$^{13}C_6$] 5-$InsP_7$, 1 μM [$^{18}O_2$] 4-$InsP_7$ (only for assignment of 4/6-$InsP_7$) and 2 μM [$^{13}C_6$] 1,5-$InsP_8$ were spiked to samples for the assignment of isomers and quantification of InsPs and PP-InsPs. 5 μL of the IS stock solution was mixed into 5 μL samples. Quantification of $InsP_8$, 5-$InsP_7$, 1-$InsP_7$, $InsP_6$, and $InsP_5$ was performed with known amounts of corresponding heavy isotopic references spiked into the samples. Quantification of 4/6-$InsP_7$ was performed with [$^{13}C_6$] 5-$InsP_7$ and Quantification of $InsP_3$ and $InsP_4$ of which no isotopic standards are available was performed with spiked [$^{13}C_6$] $InsP_6$. After spiking, 4 μM [$^{13}C_6$] 2-OH $InsP_5$, 20 μM [$^{13}C_6$] $InsP_6$, 1 μM [$^{13}C_6$] 5-$InsP_7$, 1 μM [$^{13}C_6$] 1-$InsP_7$, and 1 μM [$^{13}C_6$] 1,5-$InsP_8$ were the final concentration inside samples. All [$^{13}C_6$] inositol references [66] were kindly provided by Dorothea Fiedler.

## Yeast strains and transformation

The yeast strains were grown on YPD agar plates at 28ºC. Transformed yeast strains were grown in complete minimal medium containing the appropriate amino acids, 2% glucose, and lacking uracil to maintain selection for URA3 plasmids at 28ºC. Yeast transformations were performed by the lithium acetate method [101]. Briefly, single colony of streaked yeast strain was inoculated in 5 mL of YPD liquid medium and incubated at 28ºC in shaker incubator for overnight. Fresh 4 mL YPD liquid media was inoculated with overnight grown culture to reach $OD_{600nm}$ ~ 0.6. And culture was allowed to grow for 4 h in shaker incubator. Once the culture reached OD ~ 1, the culture was harvested at 2600 rpm for 1 min at room temperature. The pellets were washed twice with 500 μL of TE/LiAc buffer at 2600 rpm for 2 min. After final wash, the cells were resuspended with 200 μL of TE/LiAc buffer and kept on ice. These yeast competent cells were used for transformation. A total of 3.5 μL of salmon sperm DNA (approx. 8 mg/mL) was heated at 95ºC for 5 min and kept on ice for 2 min. 1 μL of plasmid (200 – 500 ng plasmid) was added to the salmon sperm DNA followed by adding 16.5 μL of yeast competent cells. The cells with plasmid were resuspended with PEG/LiAc buffer and incubated for 40 min at room temperature on rotor. After incubation, the cells were subjected to heat shock at 42ºC for 20 min. 70 μL of the cell suspension was used for plating on selection plate. The plates were incubated at 28ºC. Complementation assay was performed by dropping 8-fold serially diluted resuspension of transformed colony of yeast on selection and screening plate. S4 Table contains the list of yeast strains used in this study.

## RNA extraction, cDNA synthesis and gene expression analyses

RNA extraction and cDNA synthesis was done as described previously [45]. Briefly, 100 mg of plant tissue was used for RNA extraction with TRIzol (Sigma Aldrich) reagent, followed by DNase treatment with DNase I (NEB, M0303). A total of 2–3 µg of RNA was used for cDNA synthesis using PhiScript cDNA Synthesis Kit (dx/dt). The qPCR was performed using the DyNAmo ColorFlash SYBR Green qPCR Kit (Thermo-scientific) with CFX96 Touch Real-Time PCR Detection System (Bio-Rad Hercules, CA, USA) according to the manufacturer's protocol (Bio-Rad). Relative expression was calculated according to relative quantitation method (ΔΔCT). *PP2AA3* and Mp*ACT7* were used as reference genes for the qPCR analysis in *Arabidopsis* and *M. polymorpha*, respectively. The primers used for qPCR analyses are detailed in S1 Table.

## Extraction and HPLC analysis of inositol phosphates

Inositol polyphosphates were extracted from yeast and analyzed as described [35,102,103]. Yeast transformants were grown to midlog phase in minimal media, labelled in 2 mL of minimal media supplemented with 6 µCi/mL [³H]-*myo*-inositol (18 Ci mmol$^{-1}$; PerkinElmer). The cells were harvested, washed twice with ultra-pure water were extracted in 1 M perchloric acid extracted. For labelling of *M. polymorpha* lines, 2-week-old thalli were labelled in liquid half-strength Gamborg's B5 media supplemented with 50 µCi/mL [³H]-*myo*-inositol. After 5 days of labeling, thalli were washed two times with ultrapure water before flash frozen into liquid N2. Extracted inositol phosphates from yeast and plants were resolved by strong anion exchange high performance liquid chromatography (SAX-HPLC) using a Partisphere SAX 4.6 x 125 mm column (Whatman) at a flow rate of 0.5 mL/min with a shallow gradient formed by buffers A (1 mM EDTA) and B [1 mM EDTA and 1.3 M $(NH_4)_2HPO_4$, pH 3.8, with $H_3PO_4$] [102].

## Cloning and plasmid construction for plant transformation

Mp*IPMK* guideRNA (gRNA) designing and cloning. CRISPR/Cas9-based genome editing of Mp*IPMK* was performed as described previously [80,81]. Selection of the gRNA target site was done using CRISPRdirect web tool [104]. The gRNA protospacers were generated by annealing complementary oligonucleotides and inserted into the pMpGE_En03 vector [81] previously digested with BsaI. Mp*IPMK*-gRNA was incorporated into the binary vector pMpGE010 [81] using Gateway LR Clonase II Enzyme mix (Invitrogen, 11791100). In total, eight gRNAs were designed at different part of the gene that were used for generating CRISPR lines.

## Thallus transformation of *M. polymorpha*

Transformation of *M. polymorpha* was performed as described previously [80]. In short, 14-day-old gemmalings were sliced to eliminate the apical notches and kept for 3 days for regeneration on half strength Gamborg's B5 medium supplemented with 1% sucrose. Regenerating thalli fragments were then co-cultured with *Agrobacterium tumefaciens* GV3101 cells carrying the corresponding vectors in half strength Gamborg's B5 medium supplemented with 2% sucrose under white light and gentle shaking of 120 rpm at 22ºC. After 3 days of co-culture, the plant fragments were washed three times with sterile water and placed on half strength Gamborg's B5 medium supplemented with 100 µg/mL cefotaxime and 10 µg/mL hygromycin B.

## Construction of RNAi plasmid and generation of knockdown lines of in *A. thaliana*

Full-length coding region of *AtIPK2α* (~900 bp) was cloned into pOpOff2 (Hyg) [69] by Gateway recombination (LR ClonaseII, Invitrogen). The resulting vector pOpOFF2(Hyg)-*AtIPK2α* was used to stably transform the *atipk2β-1* knockout plants using *Agrobacterium tumefaciens* GV3101-mediated transformation by floral dipping method. Positive transformants were selected on solidified half-strength MS media containing 30 mg/mL hygromycin. Homozygous lines were identified from

selected lines at the T3 generation. For induction of RNAi, plants were treated with 25 µM dexamethasone (Dex) as indicated for each experiment.

## Thermotolerance assay

*A. thaliana* thermotolerance assays were performed as described previously [76,105–107]. For basal thermotolerance assay, 7-day-old seedlings of all the respective genotypes (Col-0, *atipk2α*, *atipk2β* and *atipk2βatipk2α^kd^*), grown on solidified half-strength MS media for 7 days, were exposed to 37ºC for 2½ days and were kept at 22ºC for recovery of another 4 days. Photographs were recorded and survival rates were counted after 4 days of recovery. For hypocotyl elongation assay, the stratified seedlings were allowed to grow at 22ºC on solidified half-strength MS media for 5 days and then transferred to 28ºC or maintained at 22ºC for another 4 days. Photographs were taken after 5 days of transfer and the hypocotyl length was measured using Image software. For basal thermotolerance assay, 7-day-old seedlings of the respective genotypes (Col-0, *atipk2α*, *atipk2β* and *atipk2βatipk2α^kd^*), grown on solidified half-strength MS media for 7 days, were exposed to 37ºC for 3 days and were kept at 22ºC for recovery of another 4 days. Photographs were recorded and survival rates were counted after 4 days of recovery.

For thermomorphogesis study in *M. polymorpha*, gemmae of wild-type and Mp*ipmk* knockout lines were transferred on half strength Gamborg's B5 media with 1% agar and grown for 5 days at 22ºC with white light of (16 h:8 h; light: dark cycle) for 5 days. On the 5th day, the plates having wild-type and Mp*ipmk* knockout lines were exposed to 37ºC in a plant chamber for 8 h. The heat-exposed plants were kept back at 22ºC. Images were taken after 13 days of recovery. Thallus area was calculated using ImageJ software. Data were analysed using GraphPad Prism 6.

## GUS staining

The 14-day-old seedlings of Col-0 and *atipk2βatipk2α^kd^* lines were fixed for 30 min in ice-cold 90% (v/v) acetone and rinsed with staining buffer (0.5 M sodium phosphate buffer pH 7.2, 10% Triton X, 10 100 mM potassium ferrocyanide, 100 mM potassium ferricyanide) without X-Gluc (500 µg ml$^{-1}$ 5-bromo-4-chloro-3-indolyl-β-D-glucuronide) followed by incubation in staining buffer supplemented with 2 mM X-Gluc at 37 °C in the dark for overnight. After a series of wash with 20%, 35% and 50% ethanol, the cleared seedlings were mounted on slide and imaged using a ZEISS Stemi 508 light microscope.

## Dual Luciferase reporter assay

A Dual Luciferase Reporter System was used to study the transient transactivation activity and was performed as described previously [76]. The promoter of *AtHSP18.1* was cloned upstream of LUC in the pGreenII 0800-LUC to generate *proAtHSP18.1:LUC* reporter. The CDS of AtHSFA1b and AtIPK2α were cloned in PEG101 vector for effector constructs. Subsequently the CDS of wild-type and catalytic dead variant of AtIPK2α were cloned in translational fusion with N-terminal GFP in pGWB652 vector. *Agrobacterium* (GV3101) strain transformants harboring the above constructs in desired combination were infiltrated in *Nicotiana benthamiana* leaves as described previously [34]. Discs of leaves were collected after 72 h of infiltration. Luciferase assay was performed by Dual-Luciferase Reporter Assay System (Promega; E1910) following manufacturer's instruction. Bioluminescence was measured using GloMax Explorer multimode microplate reader (Promega). The LUC activity was normalized to REN.

## Bimolecular fluorescence complementation assay (BiFC)

The pENTR constructs carrying CDS of *AtIPK2β*, *AtHSFA1b*, Mp*IPMK* and Mp*HSFB* were cloned in pMDC_nVENUS and pMDC_cCFP via Gateway LR recombination. *Agrobacterium* (GV3101) strain was transformed with the above constructs and different constructs were infiltrated in *N. bethamiana* leaves as described previously [34]. The lower epidermal layer was collected and mounted on slide and visualized under a confocal microscope.

## Electrophoretic mobility shift assay (EMSA)

Recombinant proteins of AtHSFA1b and MpHSF in translational fusion with the N-terminal $His_8$-MBP were expressed in *E. coli* BL21 (RIL) and subsequently lysed using PBS buffer (137 mM NaCl, 2.7 mM KCl, 10 mM $Na_2HPO_4$, 1.8 mM $KH_2PO_4$ pH 7.4) followed by Ni-NTA affinity-based purification. The heat shock element (HSE) (20 nt) was fluorescein-labelled. The FAM-labelled probes (250 nM per reaction) were incubated with purified MBP-AtHSFA1b or MBP-MpHSF with and without AtIPK2β or MpIPMK for 30 min followed by separation using 6% native PAGE. The bands were detected using Amersham ImageQuant 800 GxP biomolecular images at 460 nm.

## Yeast two hybrid assay

The full length CDS of Mp*IPMK*/*AtIPK2β* and Mp*HSF*/*AtHSFA1b* were amplified from cDNA by PCR using phusion polymerase with attB containing primers and introduced into pDONR221 (Thermo Fisher Scientific) vector using Gateway BP Clonase II Enzyme mix (Thermo Fisher Scientific). Furthermore, the coding sequences of MpIPMK and MpHSF were transferred into pGBKT7 and pGADT7, respectively, via Gateway LR recombination. Yeast transformations were performed by the lithium acetate method [101]. AH109 yeast strain was transformed with above constructs and serially spotted onto a synthetic selection media.

## Statistical analysis

Statistical analyses were performed using GraphPad Prism 6 software. The details of statistical data have been provided in S7 Table.

## Accession numbers

*AtIPK2α* (At5g07370), *AtIPK2β* (At5g61760), At*ITPK1* (At5g16760), *Actin2* (At3g18780), *AtHSP22* (At4g10250), *AtHSP70* (At3g12580), *AtHSP90* (At4g24190), *AtHSP17.6* (At5g12020), *AtHSFA1b* (At5g16820), *AtHSP18.1* (At5g59720), *PP2AA3*, *TUBULIN* (At5g62690), Mp*IPMK* (Mp1g22660.1) and Mp*ACT7* (Mp6g11010).

## Supporting information

**S1 Fig. Profile of inositol phosphates in the *M. polymorpha* thalli (Tak-1 and Tak-2), sporophylls of pteridophyta (*Nephrolepis sp*. and *Dryopteris sp*.) and seedlings of a dicot (*A. thaliana;* Col-0).** Quantification of different inositol phosphates detected in the above-mentioned embryophytes through CE-MS. Purified InsP extracts of 14-day-old *M. polymorpha* thalli (Tak-1 and Tak-2), mature sporophylls of pteridophytes (*Nephrolepis sp.* and *Dryopteris sp*.) and 14-day-old *A. thaliana* (Col-0) seedlings were subjected to CE-MS. The $InsP_5$ species were assigned by mass spectrometry and identical migration time compared with relative standards. Data are means ± SEM (n ≥ 4 biological replicates). (TIF)

**S2 Fig. AtIPK2α and AtIPK2β phosphorylates $InsP_6$ *in vitro*.** A. Structural models (overview) of AtIPK2α (Protein Data Bank entry 4FRF) (pink) and HsIP6K3 (golden). Models were obtained by the AlphaFold web portal (https://alphafold.ebi.ac.uk/) and built on the Pymol. Overlay of AtIPK2α (hot pink) and HsIP6K3 (golden) structures (RMSD value = 0.958). B and C. SDS-PAGE analyses of tag-free recombinant AtIPK2α and AtIPK2β proteins used for ITC experiments. Recombinant $His_8$-MBP-TEV-AtIPK2 proteins were subjected to TEV protease and the digested products were further purified using affinity-based chromatography. The tag-free proteins were loaded on gel. Resolved proteins were visualized by coomassie blue staining. D and E. Isothermal titration calorimetry (ITC) assays of AtIPK2α (10 μM; left panel) and AtIPK2β (10 μM; right panel) (in cell) with ITC buffer (in syringe), respectively. Raw heats per injection are shown in the top panel and the bottom panel represents the integrated heats of each injection. F and G. SDS-PAGE analysis of the recombinant AtIPK2α and its catalytic dead variants protein in translational fusion with N-terminal $His_8$-MBP-TEV tag.

Resolved proteins were visualized by coomassie blue staining (F). SDS-PAGE analysis of AtIPK2β WT and the catalytic dead variants protein. Resolved proteins were visualized by coomassie blue staining (G). H. PAGE analysis of *in vitro* kinase assay reaction products of AtIPK2s. InsP$_6$ alone and MBP served as control. I. Table showing the kinetic parameters ($K_m$ and $V_{max}$) of AtIPK2α and AtIPK2β for Ins(1,4,5)P$_3$ and InsP$_6$ at varying ATP concentration. $K_m$ and $V_{max}$ were obtained after fitting of the data against the Michaelis-Menten model. J. Quantification of the AtIPK2α reaction product using CE-MS analyses. A minor amount of 1/3-InsP$_7$ species could be detected in the reaction products. Data represent means ± SEM (n = 3). Letters depict the significance in one-way ANOVA followed by Dunnett's test (a and b, $P < 0.0001$; a and c, $P < 0.05$). K. AtIPK2β phosphorylates InsP$_6$ to synthesize 4/6-InsP$_7$ as a major PP-InsP species *in vitro*. Quantification of AtIPK2β WT and catalytic dead variant-derived reaction product using CE-MS analyses. A minor amount of 5-InsP$_7$ and 1/3-InsP$_7$ species could be detected in the reaction products. Data represent means ± SEM (n = 4). Letters depict the significance in one-way ANOVA followed by Dunnett's test (a and b, $P < 0.0001$). L. Time-dependent conversion of InsP$_6$ to 4/6-InsP$_7$, 5-InsP$_7$, and 1/3-InsP$_7$ by AtIPK2β. AtIPK2β was incubated with ATP and InsP$_6$ for different time points and the reaction products were resolved by CE-MS. Data represent means ± SEM (n = 2, replicates).
(TIF)

**S3 Fig. Generation of *atipk2α* knockdown lines in the *atipk2β-1* knockout plants.** A. Schematic diagram of the pOpOff2 vector. RB, right border; T, terminator; hyg, hygromycin; LB, left border. B. Genotyping PCR of Col-0, *atipk2β-1* and all the three *atipk2α* knockdown lines. A to E represents the primers set used for genotyping, details of the primers are mentioned S1 Table. *ACTIN* served as a reference gene. C. Representative images of the GUS signal in leaves of Col-0 and the three independent *atipk2βatipk2α^kd^* lines used in this study after dexamethasone (DEX) treatment. D. Stability of *AtIPK2α* transcript is affected in the *atipk2βatipk2α^kd^* lines. Expression analyses of *AtIPK2α* between Col-0 and independent *atipk2βatipk2α^kd^* lines using RT-PCR. *ACTIN* served as reference gene.
(TIF)

**S4 Fig. Seeds of AtIPK2-deficient lines are defective in phytic acid metabolism and heat stress induces *AtIPK2* expression.** A. PAGE analysis of seed extracts of Col-0, *atipk2βatipk2α^kd^*, *atipk2α-1* and *atipk2β-1* lines. This result is in agreement with the previously published report [36,50,63] that AtIPK2 contributes to InsP homeostasis distinctively in different plant parts. B. Quantitative RT-PCR (qRT-PCR) analysis of *AtIPK2α* and *AtIPK2β* in Col-0 after heat shock. 14-day-old seedlings were exposed to 37ºC for 3 h and were harvested for qRT-PCR analysis. *TUBULIN* was used as a reference gene. Values are means ± SEM (n = 3, biological replicates). C. CE-MS analyses of InsP extracts of Col-0 and *atipk2βatipk2α^kd^* seedlings after heat shock of 3 h at 37ºC. Graph represents the fold difference of InsP isomers of the designated genotypes upon heat stress. Values are ± SEM (n = 4, biological replicates). Statistical significance is determined in one-way ANOVA followed by Dunnett's test.
(TIF)

**S5 Fig. Characterization of the *atipk2βatipk2α*kd lines expressing *AtIPK2β* and characterization of AtIPK2α overexpression lines.** A. Photograph of the control plate maintained at 22ºC throughout basal thermal tolerance assay. This is the control set for the experiment presented in the main Fig 4C. B. Genotyping PCR of *atipk2βatipk2α^kd^* lines expressing *AtIPK2β* under the control of a constitutive 35S promoter. The primers used for genotyping are mentioned in S1 Table. *TUBULIN* served as a reference gene. C. Expression analyses of *AtIPK2β* using RT-PCR. The primers used for RT-PCR are mentioned in S1 Table. *TUBULIN* served as a reference gene. D. Genotyping PCR of *AtIPK2α in pro35S::AtIPK2α* overexpression lines. The primers used for genotyping PCR are mentioned in S1 Table. *TUBULIN* served as a reference gene.
(TIF)

**S6 Fig. AtIPK2α-type kinases are ubiquitous in green plants and share conserved catalytic motif.** A. The phylogenetic tree was estimated from an alignment of AtIPK2α amino acid sequences using maximum likelihood. Branch support

was calculated from 1000 bootstrap replicates, and values below 50% are omitted. Branch lengths are given in terms of expected numbers of amino acid substitutions per site. B. Protein alignment of MpIPMK with AtIPK2α and ScIpk2. Red rectangle marks the conserved catalytic motif PXXXDXKXG of the InsP kinase.
(TIF)

**S7 Fig. Structural model and catalytic activity of MpIPMK.** A. Cartoon depicting the conserved PXXXDXKXG motif of *M. polymorpha* IPMK. The residues highlighted in red are the altered residues, forming catalytic dead variants of IPMK, i.e., MpIPMK[D130A], MpIPMK[K132A], MpIPMK[D130AK132A] (referred as MpIPMK[DM]). B. Structural model (Structural model (overview) of MpIPMK. Models were obtained by the AlphaFold web portal (https://alphafold.ebi.ac.uk/) and built on the Pymol. C. Structural overlay of AtIPK2α (hot pink), MpIPMK (blue) structures (RMSD value = 0.872). Note the similarity between the MpIPMK model and the AtIPK2α structure (Protein Data Bank entry 4FRF). D. Zoom-in-into view of the catalytic active site of MpIPMK. E. SDS-PAGE analysis of MpIPMK and its catalytic dead variants. Arrow head denotes MpIPMK and its catalytic dead variants. F. PAGE analysis of the *in vitro* kinase reaction products of MpIPMK. Recombinant His$_8$-MBP-MpIPMK was incubated with 12.5 mM ATP, and 10 nmol InsP$_6$ at 37ºC for 12 h in reaction buffer. The reaction product was separated by 33% PAGE and visualized with toluidine blue. InsP$_6$ alone served as a control.
(TIF)

**S8 Fig. MpIPMK does not possess yeast Kcs1- or Vip1-like activity.** A. Complementation of the yeast *kcs1Δ*-associated growth defects by the ectopic expression of Mp*IPMK*. Wild-type and *kcs1Δ* yeast transformants (BY4741 background) carrying designated plasmids were spotted in 8-fold serial dilution onto YPD with and without NaCl incubated at 28ºC and 37ºC. *AtITPK1* served as positive control [30] and empty vector served as negative control. B. Complementation of *vip1Δ* -associated growth defects in yeast by ectopic expression of Mp*IPMK*. The *vip1Δ* yeast strain transformed with the episomal pCA45 (URA3) plasmids carrying Mp*IPMK* and kinase dead mutants were spotted in 8-fold serial dilutions onto uracil-free minimal medium in presence and absence of 6-azauracil. No rescue of phenotype was observed. *AtVIH2 KD* served as positive control [35] and empty vector served as negative control. C. Quantification of the reaction product of MpIPMK analyzed by CE-MS. Data represent means ± SEM (n = 2).
(TIF)

**S9 Fig. Mp*ipmk* knockout plants do not exhibit any obvious growth defects and accumulate InsP$_3$, InsP$_4$ and InsP$_5$ isomers.** A. Chromatogram showing CRISPR/Cas9-edited nucleotide sequences of Mp*ipmk1.2* and Mp*ipmk1.7* compared with those of wild-type plants using chromatogram obtained from sequencing results. B. Photograph of 14-day-old thalli of wild-type, Mp*ipmk1.2* and Mp*ipmk1.7* plants. C. CE-MS analyses of different inositol phosphates isomers in wild-type and Mp*ipmk* knockout plants. The InsP$_5$ and InsP$_7$ species were assigned by mass spectrometry and identical migration time compared with their relative standards. Two InsP$_4$ and two InsP$_3$ isomers were detected. Data are means ± SEM (n = 3, biological replicates). Inositol phosphates are represented as percentage to InsP$_6$. Significant difference is determined by one-way ANOVA followed by Dunnett's test (*$P < 0.05$, **$P = 0.002$ ***$P < 0.001$).
(TIF)

**S10 Fig. Confirmation of the transgenic *atipk2βatipk2a*kd lines expressing Mp*IPMK* heterologously under the control of a 35S promoter and qPCR analyses of the heat responsive genes between Col-0 and the *atipk2βatipk2a*kd plants.** A. Genotyping PCR of *atipk2βatipk2α$^{kd}$ lines* expressing Mp*IPMK*. The primers used for genotyping are mentioned in S1 Table. *TUBULIN* served as a reference gene. B. RT-PCR of *atipk2βatipk2α$^{kd}$ lines* expressing Mp*IPMK*. The primers used for RT-PCR are mentioned in S1 Table. *TUBULIN* served as a reference gene. C. Quantitative RT-PCR (qRT-PCR) analysis of different genes involved in thermomorphogenesis between Col-0 and the *atipk2βatipk2α$^{kd}$* lines after heat shock. 14-day-old seedlings were exposed to 37ºC for 3 h and were harvested

for qRT-PCR analysis. Transcript levels of the benchmark genes are presented relative *PP2AA3* transcript. Values are means ± SEM (n ≥ 3, biological replicates). Statistical significance is determined by two-way ANOVA followed by Tukey's test ($*P < 0.05$, $**P < 0.001$).
(TIF)

**S11 Fig. Ins(1,4,5)P$_3$, InsP$_6$ and 4/6-InsP$_7$ do not influence DNA-binding affinity of Arabidopsis heat shock transcription factor.** A and B. EMSA showing MBP, MBP-AtIPK2β do not bind directly to *HSE* element. 250 nM of the probe was used. MBP and MBP-AtIPK2β were used in the concentration ranging from 50- 200 nM. C. Ins(1,4,5)P$_3$, InsP$_6$ and 4/6-InsP$_7$ don't influence DNA-binding activity of heat shock transcription factor. 100 nM of HSF was pre-incubated with InsP$_6$, InsP$_3$ and InsP$_7$ (10 nM, 50 nM, 100 nM and 10 μM) for 30 mins followed by incubation with FAM-labelled probe for 15 mins on ice. The complexes were resolved using 6% of native PAGE.
(TIF)

**S12 Fig. MpIPMK physically interacts with MpHSFB1 and InsPs do not directly influence the transcriptional activity of MpHSFB1.** A. MpIPMK shows physical interaction with MpHSFB1 *in vivo.* AH109 yeast strain carrying the pGADT7-MpHSF and pGBKT7-MpIPMK plasmids were spotted on selective media. B. Transiently expressed MpIPMK interacts with MpHSFB1 in the nucleus of *N. benthamiana* cells. Different combination of co-expressed nVENUS and cCFP constructs were infiltrated in *N. benthamiana*. YFP represents the images taken with YFP filter and merge represents the overlay of YFP and brightfiled. Scale bar = 50 μm. C and D. EMSA showing MBP-MpIPMK do not bind directly to *HSE* element and PP-InsPs do not affect MpHSFB1 binding to *HSE* element. 250 nM of the probe was used. MpIPMK was used in the concentration ranging from 50- 200 nM. Ins(1,4,5)P$_3$, InsP$_6$ and 4/6-InsP$_7$ don't influence DNA-binding activity of heat shock transcription factor. 100 nM of HSFB1 was pre-incubated with InsP$_6$, InsP$_3$ and InsP$_7$ (10 nM, 50 nM, 100 nM and 10 μM) for 30 mins followed by incubation with FAM-labelled probe for 15 mins on ice. The complexes were resolved using 6% of native PAGE.
(TIF)

**S1 Table. List of primers used in the study.**
(XLSX)

**S2 Table. List of hits from Phyre[2] analyses using Human IP6K3.**
(XLSX)

**S3 Table. AtIPK2α and AtIPK2β expression analysis under heat stress.**
(XLSX)

**S4 Table. Yeast Strains used in this study.**
(XLSX)

**S5 Table. Sequences used in Phylogenetic analysis.**
(XLSX)

**S6 Table. MS parameters for MRM transitions.**
(XLSX)

**S7 Table. Details of statistical analysis.**
(XLSX)

**S8 Table. Raw data file.**
(XLSX)

## Acknowledgments

We acknowledge Cristina Azevedo for the pCA45 plasmid. We acknowledge Sandeep M. Eswarappa for their Luminometer facility. We are grateful to Manoj Majee for the pGreenII 0800-LUC and pEarlyGate101 vectors. We thank Saikat Bhattacharjee for providing us the Arabidopsis *ipk2a-1* seeds. We thank Utpal Nath for the AH109 yeast strain. We thank all members of the Laha Lab for their critical feedback to this study.

## Author contributions

**Conceptualization:** Debabrata Laha.

**Formal analysis:** Ranjana Yadav, Guizhen Liu, Henning J. Jessen, Debabrata Laha.

**Funding acquisition:** Debabrata Laha.

**Investigation:** Ranjana Yadav, Guizhen Liu, Priyanshi Rana, Naga Jyothi Pullagurla, Danye Qiu.

**Methodology:** Ranjana Yadav, Guizhen Liu, Priyanshi Rana, Danye Qiu.

**Project administration:** Debabrata Laha.

**Resources:** Henning J. Jessen, Debabrata Laha.

**Supervision:** Henning J. Jessen, Debabrata Laha.

**Validation:** Ranjana Yadav, Guizhen Liu, Priyanshi Rana, Naga Jyothi Pullagurla, Danye Qiu, Henning J. Jessen, Debabrata Laha.

**Visualization:** Ranjana Yadav, Priyanshi Rana, Naga Jyothi Pullagurla, Debabrata Laha.

**Writing – original draft:** Ranjana Yadav, Debabrata Laha.

**Writing – review & editing:** Ranjana Yadav, Guizhen Liu, Henning J. Jessen, Debabrata Laha.

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
