## [Decision Letter · Decision Letter 0]

25 Feb 2025

PGENETICS-D-24-01385

Conservation of heat stress acclimation by the IPK2-type kinases that control the synthesis of the inositol pyrophosphate 4/6-InsP7 in land plants

PLOS Genetics

Dear Dr. Laha,

Thank you for submitting your manuscript to PLOS Genetics. After careful consideration, we feel that it has merit but does not fully meet PLOS Genetics's publication criteria as it currently stands. Therefore, we invite you to submit a revised version of the manuscript that addresses the points raised during the review process.

Please submit your revised manuscript within 60 days (April 26, 2025). If you will need more time than this to complete your revisions, please reply to this message or contact the journal office at plosgenetics@plos.org. Please include the following items when submitting your revised manuscript:

We look forward to receiving your revised manuscript.

Kind regards,

Bao-Cai Tan

Academic Editor

PLOS Genetics

Bao-Cai Tan

Academic Editor

PLOS Genetics

Aimée Dudley

Editor-in-Chief

PLOS Genetics

Anne Goriely

Editor-in-Chief

PLOS Genetics

**Additional Editor Comments (if provided):**

Dear Dr. Laha,

I apologize for the delay in reviewing this manuscript, as we have difficulty securing reviewers. As you may see from the review comments, both reviewers find the identification of IPK2s as inositol pyrophosphate synthesizing enzymes and their involvement in heat stress very interesting. I think that it is a new finding that advances the function of inositol pyrophosphate 4/6-InsP7 in plants. However, the reviewers also raised serious questions on data interpretation, experiment design (proper controls used), and the sufficiency of evidence for the conclusion. I agree with the reviewer that necessary controls should be used in the ITC experiments and the EMSA assays in order to draw a convincing conclusion. The IP3/4/5 kinase activities in this manuscript should be compared with these previously reported. In addition, how AtIPK2α enhances the DNA-binding and transcription activity of AtHSFA1b should be addressed. In light of these concerns, I recommend a major revision. I would encourage a resubmission after the questions and concerns are addressed.

**Journal Requirements:**

**Reviewers' comments:**

Reviewer's Responses to Questions

**Comments to the Authors:**

Reviewer #1: The authors conducted CE-MS analysis and identified that 4/6-InsP7 is the predominant PP-InsP species present in Arabidopsis and liverworts. They also identified the enzymes responsible for the biosynthesis of 4/6-InsP7 and further tested the evolutionary conservation of these enzymes in generating 4/6-InsP7 and regulating heat stress tolerance. The findings are quite interesting, and I have a few comments.

1. Line 118: The authors state, “To investigate whether 4/6-InsP7 is ubiquitous across land plants.” However, in the manuscript, they only examine the inositol phosphate profiles from liverworts and Arabidopsis, which seems insufficient to support the claim of “land plants.” It would be better to include several land plants of evolutionary significance or cite relevant literature to substantiate this claim.

2. If possible, it would be beneficial to generate an atipk2β(kd)atipk2α mutant to further support the conclusion that AtIPK2α and AtIPK2β act redundantly to regulate the synthesis of 4/6-InsP7 in Arabidopsis.

3. In Supplementary Data Set 3, which column represents the transcripts of IPK2 under heat stress? The authors need to label the treatments clearly.

4. Lines 342-355: The authors utilize the ipk2∆ knockout strain to test the functional conservation among yeast, M. polymorpha, and Arabidopsis. What is known about the role of IPK2 in yeast? The authors need to introduce the function of yeast IPK2 in the introduction.

5. I am curious whether the overexpression (OE) of IPK2 could significantly enhance the heat tolerance of Arabidopsis plants. The manuscript only discusses the phenotypes of complementary lines and does not mention the OE lines.

6. The authors performed EMSA assays and concluded that the AtIPK2α protein enhances the DNA-binding and transcription activity of AtHSFA1b. Is the IPK2 protein localized in the nucleus? Does AtIPK2α interact with AtHSFA1b at the protein level? BiFC, split-luciferase, or Co-IP experiments would be necessary to validate this interaction. The incubation of AtHSFA1b with Ins(1,4,5)P3, InsP6, and 4-InsP7 did not affect the DNA-binding activity of AtHSFA1b. Therefore, AtIPK2 appears to have biological functions in catalyzing the formation of 4/6-InsP7 and enhancing the DNA-binding and transcription activity of AtHSFA1b. Do these two functions occur in the same subcellular location, such as the nucleus? Is AtIPK2 localized in both the cytosol and the nucleus? The authors need to address these questions.

Reviewer #2: The manuscript by Yadav and colleagues describes the identification of IPK2s from the model plants Arabidopsis and Marchantia as inositol pyrophosphate synthesizing enzymes involved in heat stress adaptation. The manuscript reports a number of interesting findings including new mutant alleles for IPK2s and a strong genetic connection between IPK2 function and plant heat stress responses. However, it is unclear of indeed InsP6 is the best substrate for IPK2s and if the enzyme directly regulates cellular 4/6-InsP7 levels as well as the transcriptional activity of plant heat shock transcription factors. I summarize my comments and suggestions below:

1. MINOR Introduction, Line 77-78. There is, to the best of my knowledge, no strong evidence that plant PPIP5Ks are InsP6 kinases. A Mpvip1ge mutant in Marchantia polymorpha has wild-type like levels of 1-InsP7, increased levels of 5-InsP7 and decreased levels of 1,5-InsP8 (https://pubmed.ncbi.nlm.nih.gov/39531477/). This suggests that at least in planta, Vip1 is not an InsP6 kinase, but may use exclusively use 5-InsP7 as substrate to generate 1,5-InsP8.

2. MINOR Line 92-94. 4/6-InsP7 levels have been reported for Arabidopsis seedling and for Marchantia Tak-1 plants here as well https://pubmed.ncbi.nlm.nih.gov/39531477/.

3. MAJOR Line 126-127. 4/6-InsP7 levels have been previously determined in Arabidopsis (references cited in the introduction + https://pubmed.ncbi.nlm.nih.gov/39531477/, see above) seedlings and in Marchantia polymorpha Tak-1 and various mutants in the PP-InsP biosynthesis and catabolic pathway (https://pubmed.ncbi.nlm.nih.gov/39531477/). In this previous study, both in Arabidopsis and in Tak-1 4/6-InsP7 levels are not significantly higher compared to either 1-InsP7, 5-InsP7 or 1,5-InsP8 (compare Fig. 1F and 3G in https://pubmed.ncbi.nlm.nih.gov/39531477/, respectively). Given that the absolute abundance of a certain PP-InsP species is poorly correlated with its biological signaling capacity, I would consider removing the statement that 4,6-InsP7 is "the most abundant InsP7 isomer". It would also be helpful if the authors could plot the total concentrations of InsPs and PP-InsPs per g of tissue weight, rather than as a fraction of InsP6 levels.

5. MINOR A structural superposition comparing HsIP6K3 and AtIPK2 in Ca representation should be presented, with corresponding structural alignment statistics (root mean square deviation in Angstroems comparing X corresponding Calpha atoms matching between the two structures) in revised Figure S6C.

6. MAJOR: Lines 168-180 and Figure 1C. The ITC experiments should to be repeated with appropriate controls. It is not sufficient to report the raw heats from injections of substrate (in the syringe) into the enzyme (in the cell). Essential control experiments are 1) injection of substrate (in the syringe) into buffer (in the cell, 50 mM Tris Cl pH 7.0 and 0.5 mM EDTA) and 2) injection of buffer (in the syringe, 50 mM Tris Cl pH 7.0 and 0.5 mM EDTA) into enzyme (in the cell). The raw heats of these control experiments should be plotted alongside the actual experiment in the upper panels of Figure 1C. Titration of InsPs into free buffer is usually associated with large DPs, this needs to be checked and corrected for. Binding stoichiometries (N) should not be fixed during data fitting and should be reported alongside the derived dissociation constants. It is presently hard to understand how, for example, the raw heats in the top left experiment in panel 1C could be fitted at to the binding curve. The method section should include the injection volumes for each injection and the method of fitting. How was the pH of the ligand in ITC buffer dilutions adjusted? Why is the Kd derived by ITC (0.5 micromolar) drastically different from the Km estimate (~250 micromolar)?

7. MAJOR The newly reported kinase activity for IPK2a needs to be compared to the well established IP3/4/5 kinase activities previously reported for this enzyme. AtIPK2a has been previously characterized as an InsP3 kinase, which can also accept InsP4 and InsP5 as potential substrates (https://pubmed.ncbi.nlm.nih.gov/12226109/) Specifically, AtIPK2a has been characterized to phosphorylate 1,4,5-InsP3 to 1,4,5,6-InsP6, or 1,3,4,5-InsP4 to 1,3,4,5,6-InsP5 i.e. phosphorylating the 6-position of the inositol ring, with Km's in the low micromolar range, and with Vmax of ~ 100 nmol min-1 mg-1. In contrast, the Km for InsP6 is 250 micromolar, with a Vmax of only 2 nmol min-1 mg. This suggests that InsP6 is a very poor substrate for AtIPK2, when compared to InsP3 or InsP4 (https://pubmed.ncbi.nlm.nih.gov/12226109/). The authors should consider performing comparative enzyme kinetics with AtIPK2a and InsP3, InsP4 and InsP6 as substrate to define the best in vitro substrate for this enzyme. Based on the current data the statement line 216-219 is not supported by robust experimentation.

8. MAJOR: Figure 2C The atipk2b/atipk2a knock-down line contains higher levels of InsP3, InsP4 and InsP5 but only very small variations in 4/6-InsP7. How come the difference in for example InsP5 levels between Col-0 (~1.6 pmol mg-1 FW) and atipk2b/atipk2akd line2 (~3 pmol mg-1 FW) are labeled statistical insignificant, while in the case of 4/6-InsP7 much smaller differences (Col-0 ~0.6 pmol mg-1 FW vs. 0.4 pmol mg-1 FW) are labeled as highly significant? Why are the total levels of InsPs/PP-InsPs so different for the Col-0 control when comparing the experiments shown in Figure 2A and 2D)? The authors should provide further details on the statistical tests used and discuss the possibility that the altered InsP3/4/5 levels may indirectly affect 4/5-InsP7 concentrations rather than the IPK2 enzymes themselves (see above, point #7).

10. MAJOR 283-297 and Figure 3A. Consistent with Figure 2C, 4/6-InsP7 levels are reduced in Figure 3A when comparing the Col-0 wild type with the atipk2b/atipk2a knock-down lines. During heat stress, InsP4/6 levels are reduced in Col-0 and in the mutant lines, which to me would suggest that heat stress-induced changes of 4/6-InsP7 levels occur independent of atipk2a and atipk2b. Can the authors explain why (line 296-297) :"Collectively, our data suggests that AtIPK2α and AtIPK2β function together to control cellular levels of 4/6-InsP7 during heat stress in planta." ?

11. MAJOR: Figure 5D, as outlined above it would be worthwhile to compare the InsP6 kinase activity of MpIPMK with its ability to phosphorylate InP3 and InsP4.

12. MAJOR: Figure 6B. Would the accumulation of another InsP3 peak in the Mpipmk knock-out line not argue for this InsP3 being a preferred in vivo substrate for MpIPMK? In line with this, InsP3 levels are up in Supplementary Figure 8C, again indicating that the lower 4/6-InsP7 levels in this mutant could be caused by a missing InsP4 precursor required for 4/6-InsP7 biosynthesis.

13. MINOR: Figure 8A. Based on what criteria were the heat shock response target genes chosen for this analysis?

14. MINOR: Figure 8B. Is it known that HSFA1b is regulating expression of HSP70, HSP22, HSP101, HSP17.6, HSP17.4, HSP18.1 and HSFA2 or how was this particular transcription factor selected? Also, there are many possible molecular scenarios that could explain why expression of AtIPK2a may lead to enhanced transcriptional activity of AtHSFA1b. What is the rational to focus on the direct interaction of the enzyme with the transcription factor, a rather unusual mode of transcription factor regulation? Have other scenarios been tested experimentally?

15. MAJOR: Figure 9 all panels. All gels are not very well resolved (protein-DNA complexes too close to the wells). Consider using a lower percentage of acrylamide to better separate the different molecular species. The EMSA assays require additional controls: Currently, there is no competitor DNA in the EMSA reaction. With the current experimental settings, any positively-charged DBD will bind to the negatively charged probe. Consider including a control experiment with a competitor DNA (like salmon sperm DNA) to establish the binding specificity of the probe. Along these lines, a mutated probe that abolishes binding should be tested alongside. Report molar concentrations for all proteins and probes concentrations. The current figure legend provides an interpretation rather than a description of the experiment. Line 459 change 'a short nucleotide fragment' for 'a short double stranded oligonucletotide'. Figure 9B, the enhancement of DNA binding in Figure 9B could be due to an artifact. Use another enzyme to check that the observed effect is specific to IPK2a (or to a domain of IPK2). Also, which amount of TF is used in Figure 9B? The legend for Figure 9D is incomplete. Figure 9E, If the hypothesis is that the enzyme affects the TF binding by interacting with it, then there should be a band for the TF alone and an upper band corresponding to the TF + AtIPK2a complex binding to the probe. But there seems to be only one band? The best way to address this point would be to identity missense mutations in the enzyme that impair the interaction of this mutant version with the TF and the probe in EMSA assays. The enhancement of DNA-binding in Figure 9E is not convincing. If the AtIPK2a enzyme indeed promotes DNA binding, then the intensity of the free probe should at least decrease. To aid comparison of Figure 9D and Figure 9E, the reactions should be performed with the same amounts of the HSF alone + probe (lane2). The percentage of bound probe should be similar but it is clearly not the same. For consistency, the interaction between the enzyme and the TF should also be tested for the Arabidopsis proteins, not only for the Marchantia orthologs in the Y2H assays, and possibly a second method (co-IP, in vitro pull down) could be used to substantiate this interaction.

**Have all data underlying the figures and results presented in the manuscript been provided?**

Reviewer #1: Yes

Reviewer #2: Yes

PLOS authors have the option to publish the peer review history of their article (what does this mean? ). If published, this will include your full peer review and any attached files.

**Do you want your identity to be public for this peer review?** For information about this choice, including consent withdrawal, please see our Privacy Policy .

Reviewer #1: No

Reviewer #2: No

**Figure resubmission:**
---

## [Decision Letter · Decision Letter 1]

22 Jul 2025

PGENETICS-D-24-01385R1

Conservation of heat stress acclimation by the IPK2-type kinases that control the synthesis of the inositol pyrophosphate 4/6-InsP7 in land plants

PLOS Genetics

Dear Dr. Laha,

The revised manuscript has undergone substantial improvements. All concerns have been addressed, except for a few minor ones raised by Reviewer 1. I am pleased to accept this manuscript for publication pending a minor revision. Thanks for submitting your best work to PLOS Genetics.

Please submit your revised manuscript within 30 days. If you will need more time than this to complete your revisions, please reply to this message or contact the journal office at plosgenetics@plos.org. Please include the following items when submitting your revised manuscript:

We look forward to receiving your revised manuscript.

Kind regards,

Bao-Cai Tan

Academic Editor

PLOS Genetics

Aimée Dudley

Editor-in-Chief

PLOS Genetics

Anne Goriely

Editor-in-Chief

PLOS Genetics

**Journal Requirements:**

**Reviewers' comments:**

Reviewer's Responses to Questions

**Comments to the Authors:**

Reviewer #1: All my questions have been addressed

Reviewer #2: The revised manuscript by Yadav et al. represents a substantial improvement and now more accurately reflects the experimental work presented. The EMSA assays have been refined, and the discussion surrounding the InsP₃ versus InsP₆ kinase activity has become more balanced.

Figure 1C: The ITC experiments are now supported by appropriate control experiments. However, the observed stoichiometries do not align with a simple 1:1 enzyme–substrate model, likely due to the small differential power signal and considerable data noise. While this does not undermine the experiment, it suggests that the derived dissociation constants may lack accuracy. I recommend leaving the experiments as they are.

Lines 198–200: The current reasoning is convoluted — implying that low activity toward a substrate confirms its correctness is not convincing. It would be more straightforward to state that AtIPK2 is a relatively inefficient InsP₆ kinase, and that InsP₃ appears to be the preferred in vitro substrate.

**Have all data underlying the figures and results presented in the manuscript been provided?**

Reviewer #1: Yes

Reviewer #2: Yes

PLOS authors have the option to publish the peer review history of their article (what does this mean? ). If published, this will include your full peer review and any attached files.

**Do you want your identity to be public for this peer review?** For information about this choice, including consent withdrawal, please see our Privacy Policy .

Reviewer #1: No

Reviewer #2: No

**Figure resubmission:**
---

## [Editor Report · Decision Letter 2]

13 Aug 2025

Dear Dr. Laha,

We are pleased to inform you that your manuscript entitled "Conservation of heat stress acclimation by the IPK2-type kinases that control the synthesis of the inositol pyrophosphate 4/6-InsP7 in land plants" has been editorially accepted for publication in PLOS Genetics. Congratulations!

Yours sincerely,

Bao-Cai Tan

Academic Editor

PLOS Genetics

Bao-Cai Tan

Academic Editor

PLOS Genetics

Aimée Dudley

Editor-in-Chief

PLOS Genetics

Anne Goriely

Editor-in-Chief

PLOS Genetics

Comments from the reviewers (if applicable):

**Data Deposition**

http://datadryad.org/submit?journalID=pgenetics&manu=PGENETICS-D-24-01385R2

**Press Queries**

---

## [Editor Report · Acceptance letter]

PGENETICS-D-24-01385R2

Conservation of heat stress acclimation by the IPK2-type kinases that control the synthesis of the inositol pyrophosphate 4/6-InsP7 in land plants

Dear Dr Laha,

We are pleased to inform you that your manuscript entitled " 

Conservation of heat stress acclimation by the IPK2-type kinases that control the synthesis of the inositol pyrophosphate 4/6-InsP7 in land plants" has been formally accepted for publication in PLOS Genetics! Your manuscript is now with our production department and you will be notified of the publication date in due course.

With kind regards,

Anita Estes

PLOS Genetics

On behalf of:
